# CROSS-MODAL REDUNDANCY AND THE GEOMETRY OF VISION–LANGUAGE EMBEDDINGS

**Grégoire Dhimoïla**[a,b,c]  **Thomas Fel**[d]  **Victor Boutin**[e]  **Agustin Picard**[c]

[a]Brown University  [b]ENS Paris Saclay  [c]DEEL - IRT Saint Exupéry
[d]Kempner Institute, Harvard University  [e]CNRS

⬡ https://github.com/Parabrele/IsoEnergy

## ABSTRACT

Vision–language models (VLMs) align images and text with remarkable success, yet the geometry of their shared embedding space remains poorly understood. To probe this geometry, we begin from the Iso-Energy Assumption, which exploits cross-modal redundancy: a concept that is truly shared should exhibit the same average energy across modalities. We operationalize this assumption with an Aligned Sparse Autoencoder (SAE) that encourages energy consistency during training while preserving reconstruction. We find that this inductive bias changes the SAE solution without harming reconstruction, giving us a representation that serves as a tool for geometric analysis. Sanity checks on controlled data with known ground truth confirm that alignment improves when Iso-Energy holds and remains neutral when it does not. Applied to foundational VLMs, our framework reveals a clear structure with practical consequences: (*i*) sparse *bimodal* atoms carry the entire *cross-modal* alignment signal; (*ii*) *unimodal* atoms act as *modality-specific* biases and fully explain the modality gap; (*iii*) removing *unimodal* atoms collapses the gap without harming performance; (*iv*) restricting vector arithmetic to the *bimodal* subspace yields in-distribution edits and improved retrieval. These findings suggest that the right inductive bias can both preserve model fidelity and render the latent geometry interpretable and actionable.

## 1 INTRODUCTION

Vision-language models (VLMs) (Radford et al., 2021; Zhai et al., 2023; Fini et al., 2025; Tschannen et al., 2025) have become central to applications from visual question answering (Chen et al., 2024) and medical imaging (Singhal et al., 2023) to autonomous driving (Zhou et al., 2024) and embodied AI (Shukor et al., 2025), creating shared embedding spaces where visual and textual representations of similar concepts align. Despite their empirical success, we lack a principled understanding of how these models internally organize and align semantic content across modalities. This work investigates the geometric structure of cross-modal embeddings: *How do VLMs organize the alignment between visual and textual semantics, and what principles govern this shared representational space?* Understanding these mechanisms is crucial for designing more robust and interpretable vision-language architectures.

**From Attributions to Concepts.** To address this, the interpretability community has developed an array of tools (Gilpin et al., 2018; Bau et al., 2017) aimed at dissecting learned representations. Early efforts centered on attribution methods (Zeiler & Fergus, 2014; Sundararajan et al., 2017; Smilkov et al., 2017; Petsiuk et al., 2018; Fel et al., 2021) that highlight "where" a model focuses its attention, but these approaches often fall short (Nguyen et al., 2021; Kim et al., 2022; Colin et al., 2022; Hase & Bansal, 2020; Sixt et al., 2022) of explaining "what" abstractions and on which basis the model operates. More recent concept-based methods (Ghorbani et al., 2019; Zhang et al., 2021; Fel et al., 2023b; Elhage et al., 2022; Fel et al., 2023a) have emerged to answer this, seeking to extract meaningful features (concepts) that models implicitly compute over. Concept extraction is typically

---

*Correspondence to:* gregoire.dhimoila@ens-paris-saclay.fr

framed as a dictionary learning problem (Tošić & Frossard, 2011; Rubinstein et al., 2010; Elad, 2010; Mairal et al., 2014; Dumitrescu & Irofti, 2018): identifying an overcomplete set of basis vectors that explain internal activations via sparse coding (Olshausen & Field, 1996; 1997; Lee et al., 2006; Foldiak & Endres, 2008; Rentzeperis et al., 2023). This approach is a direct response to an underlying phenomenology of the activation space: the Linear Representation Hypothesis (LRH) (Elhage et al., 2022; Wattenberg & Viégas, 2024) which posits that model activations can be viewed as sparse combinations of latent directions drawn from a high-dimensional concept basis. This is motivated not only by empirical findings, but also by geometric arguments: in high-dimensional spaces, sparse sets of nearly orthogonal directions can represent exponentially many distinct concepts with minimal interference, a principle reminiscent of Johnson-Lindenstrauss-style (Johnson et al., 1984; Larsen & Nelson, 2017) embeddings. Sparse autoencoders (SAEs) (Makhzani & Frey, 2014; Elhage et al., 2022) directly operationalize this phenomenology by learning an approximate inverse mapping from model activations to latent conceptual directions. They recover a sparse code that identifies which overcomplete basis elements (concepts) are active in a given representation, and have been effective at uncovering semantically meaningful structure in both vision (Gorton, 2024; Fel et al., 2025; Dreyer et al., 2025; Rao et al., 2024) and language models (Cunningham et al., 2023; Bricken et al., 2023; Rajamanoharan et al., 2024; Gao et al., 2025; Surkov et al., 2025).

**Multimodal Interpretability.** However, applying SAEs to VLMs (Bhalla et al., 2024; Pach et al., 2025) reveals a puzzling behavior: concept dictionaries often segregate by modality (Papadimitriou et al., 2025). Although VLMs are trained for cross-modal alignment, the extracted concepts tend to activate exclusively for either image or text inputs. This observation is the concept-based view of a now well-documented phenomenon called the *modality gap*. Prior work has described this separation geometrically, attributing it to a conical structure in the embedding space (Liang et al., 2022; Ethayarajh, 2019), or through the lens of training dynamics induced by the contrastive loss (Fahim et al., 2024; Shi et al., 2023; Yaras et al., 2024). Yet these accounts do not explain what such separation means at the level of shared concepts.

In this work, we introduce a framework for analyzing multimodal representations grounded in an explicit generative model. Central to this framework is the *Iso-Energy Assumption* – if a concept is truly shared across modalities, it should exhibit invariant energy, defined as the average squared activation, regardless of input domain. This assumption provides a concrete, testable criterion for identifying *bimodal* concepts. We then operationalize this simple assumption with a natural method: an alignment-penalized Matching Pursuit Sparse Autoencoder (Aligned SAE), which encourages energy consistency across modalities during training. This approach allows us to diagnose whether extracted concepts genuinely support cross-modal alignment or merely reflect modality-specific patterns. This work makes the following contributions:

- We introduce the *Iso-Energy Assumption*, which exploits cross-modal redundancy by requiring that shared concepts exhibit the same average activation energy in image and text.
- We operationalize this assumption with an Aligned Sparse Autoencoder that enforces energy consistency during training while preserving reconstruction, and validate it with ground-truth sanity checks where classical SAEs fail.
- Applied to dual-encoder vision–language foundation models, this inductive bias reveals a geometric decomposition invisible to classical SAEs: (***i***) sparse *bimodal* atoms carry the entire *cross-modal* alignment signal, while (***ii***) *unimodal* atoms carry *modality-specific* information and fully explain the modality gap, with a few high energy atoms acting as *modality-specific* biases.
- Moreover, our work reveals that cross-modal information is carried by shared atoms, as opposed to idiosyncratic ones described by Papadimitriou et al. (2025).
- This structure enables actionable interventions without loss of performance: (***iii***) removing *unimodal* atoms collapses the modality gap, and (***iv***) restricting vector arithmetic to the *bimodal* subspace yields in-distribution edits.

At the core lies a single theoretical premise: if concepts are genuinely shared across modalities, they must imprint redundant statistical traces in each domain. We formalize this intuition by modeling multimodal representations as partial inverses of a shared generative process, and introducing the *Iso-Energy Assumption* as the inductive bias that makes concept recovery feasible.

**Nomenclature.** We distinguish the following terms when referring to concepts based on their behavior and based on the type of information they carry. (***i***) ***Activation patterns:*** *(a) unimodal* concepts activate exclusively on a single modality, whereas *(b) bimodal* concepts activate on both. This

distinction is made by thresholding the modality score $\mu$, a comparison of domain-wise energy (see Appendix E.2.2). ***(ii) Information carried:*** *(c) modality-specific* concepts carry information specific to a given modality, e.g. that an image has cropping artifacts, while *(d) cross-modal* concepts carry cross-modal information and participate in the contrastive geometric alignment between modalities.

## 2    RELATED WORK

Previous work extensively describes salient phenomena in the shared space of multimodal dual-encoders (Schrodi et al., 2025; Levi & Gilboa, 2025; Jiang et al., 2023; Udandarao, 2022). Most notably, Liang et al. (2022) describes what is now commonly known as the *modality-gap*: that image and text embeddings reside in disjoint cones in the latent space. This modality gap has been attributed to a cone effect (Liang et al., 2022; Ethayarajh, 2019; Schrodi et al., 2025) and training dynamics induced by the contrastive loss and by mismatched pairs of data (Fahim et al., 2024; Shi et al., 2023; Yaras et al., 2024). (Levi & Gilboa, 2025) additionally shows that embeddings are contained near the surface of ellipsoid empty shells centered near the mean of the two distributions $\mu_I$ and $\mu_T$.

The *cone effect* naturally comes with a salient difference in modality wise means $\Delta := \mu_I - \mu_T$ (Levi & Gilboa, 2025; Liang et al., 2022; Fahim et al., 2024), or a perfect linear separability between image and text embeddings (Schrodi et al., 2025; Levi & Gilboa, 2025; Fahim et al., 2024; Shi et al., 2023). Furthermore, Schrodi et al. (2025) shows that, surprisingly, a small subset of coordinates in the canonical latent basis accounts for most of the norm of $\Delta$. Previous work tries to get rid of the modality gap by shifting modality-wise means (Liang et al., 2022; Levi & Gilboa, 2025), or by projecting out the few canonical directions mentioned above (Schrodi et al., 2025). In all of these works, the proposed intervention decreases cross-modal performance of embeddings, with the notable exception of Zhang et al. (2023), whose Proposition 1 precedes our more general Proposition 1. Figure 4 shows that this difference in means, while explaining the bulk of the modality gap and linear separability between the two sets of embeddings, is not enough to explain the full distributional mismatch. For that, we need to account for modality-specific information.

Jiang et al. (2023) takes an information-theoretic angle to show that modality specific variability in representations is necessary to preserve unimodal capabilities. They use this insight to design a new architecture that would explicitly regularise for the representation of such information. We show that *(i)* unregularised foundational VLM encoders possess such representations, and that *(ii)* the underlying mechanism is to organise these types of information linearly in distinct subspaces—let us call $\Omega_I$ and $\Omega_T$ the subspaces containing modality specific information, and $\Gamma$ the one containing shared information. The projection by Schrodi et al. (2025) described above, though not introduced for this purpose, is a first attempt at identifying and intervening on $\Omega_I \oplus \Omega_T$, characterised by the span of the canonical basis vector selected, while $\Gamma$ would be the orthogonal complement. However, their results are negative, as their intervention is detrimental to model performance even on cross-modal focused tasks—indicating significant unintended alteration of $\Gamma$—while leaving the gap wide open (see Appendix L), thus showing that their identified directions can't describe $\Omega_I \oplus \Omega_T$.

Our novelty lies not in the description of most phenomena discussed above, but rather in the characterisation of these structures in foundational VLM encoders. Through the use of a concept-based approach, we are able to identify *(i)* high-energy unimodal features to unimodal biases, *(ii)* $\Gamma = \mathrm{Cone}(\boldsymbol{\delta} \cdot \boldsymbol{D})$ and $\Omega_{I/T} = \mathrm{Cone}(\boldsymbol{\delta}_{I/T} \cdot \boldsymbol{D})$. Here, $\mathrm{Cone}(\boldsymbol{e}_1, \ldots, \boldsymbol{e}_k) = \{\sum_{i=1}^{k} \lambda_i \boldsymbol{e}_i \mid \lambda_i \geq 0\}$, $\boldsymbol{D} \in \mathbb{R}^{K \times d}$ is the set of $K$ dictionary atoms of latent dimension $d$, and $\boldsymbol{\delta}$ (resp. $\boldsymbol{\delta}_I, \boldsymbol{\delta}_T) \in \{0, 1\}^d$ is the binary mask selecting bimodal (resp. image-, text-only) atoms. To claim this characterisation, we carefully analyse four complementary aspects of the dictionary through novel metrics. Once validated, we further test its practical value through targeted interventions on these structures.

## 3    EXPLOITING CROSS-MODAL REDUNDANCY FOR CONCEPT RECOVERY

We formalize the setting by modeling each datum (image, caption, ...) as generated from a latent concept vector and a domain-specific generator. From this perspective, encoders serve as partial inverses of a shared multimodal generative process. However, recovering the underlying concepts is in general an ill-posed inverse problem: without additional inductive biases, nonlinear ICA is provably unidentifiable (Hyvarinen & Morioka, 2016; Locatello et al., 2019; Khemakhem et al., 2020)—infinitely many mappings can account for the observed data in the absence of cross-domain constraints. Introducing Iso-Energy provides a selection principle among these solutions and yields a representation that is useful for studying the geometry observed in practice. We begin by formally

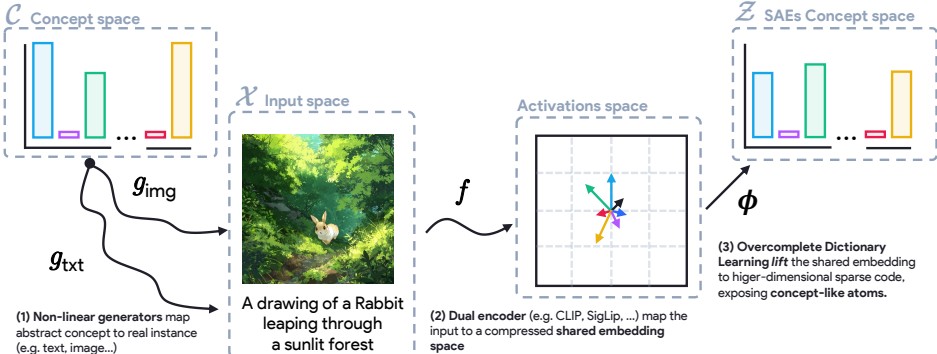

Figure 1: **Multimodal data-generating process.** A latent concept vector $c \in \mathcal{C}$ (e.g., *rabbit*, *forest*, *light*, *running*) is sampled as a sparse combination of abstract concepts and rendered through domain-specific generators $g(\cdot)$ (e.g., image or text). Dual-encoder models (e.g., CLIP, SigLIP) map these observations to a shared activation space, which sparse autoencoders (or other overcomplete dictionary learning methods) then attempt to *lift* back to concept-like atoms. However, without additional inductive bias, encoder-decoder pairs $(f, \phi)$ are not uniquely determined, a well-known identifiability problem in nonlinear ICA. Here we leverage cross-modal redundancy as a useful inductive bias, nudging the solution toward recovering *bimodal* concepts.

introducing our data generative process: each datum arises by (*i*) sampling a sparse concept vector and (*ii*) rendering it through a domain-specific generator. Formally, let $K \in \mathbb{N}$ be the number of latent concepts and $\mathcal{C} \subseteq \mathbb{R}^K$ the concept space. Let $\mathfrak{D} = \{d_1, \dots, d_{|\mathfrak{D}|}\}$ index domains with observation spaces $\mathcal{X}^{(d)}$.

**Definition 1** (Multimodal Concept Generative Process). *Sample $c \sim p = \prod_{k=1}^K p_k$ with $|\mathrm{supp}(c)| \ll K$. For each $d \in \mathfrak{D}$, a deterministic generator $g^{(d)} : \mathcal{C} \to \mathcal{X}^{(d)}$ yields $x^{(d)} = g^{(d)}(c)$. We admit that each $g^{(d)}$ is $C^1$ and locally invertible.*

A VLM encoder $f$ maps each observation $x^{(d)}$ to a shared embedding, and a sparse autoencoder $\phi$ attempts to *disentangle* this embedding back into concept coordinates, so that $\phi \circ f$ approximates the inverse of Definition 1. However, recovering latent concepts from nonlinear generators is not identifiable in general: without additional structure, many different dictionaries can explain the same data (Hyvarinen & Morioka, 2016; Khemakhem et al., 2020; Locatello et al., 2019). This ambiguity is visible in practice, where SAE dictionaries often vary substantially across random seeds (Fel et al., 2025). Domain membership provides auxiliary information, but by itself it is insufficient to ensure consistent recovery. We therefore introduce a cross-domain simple inductive bias, the *Iso-Energy Assumption*, which states that genuinely multimodal concepts should maintain consistent average energy across modalities. Because such concepts manifest in parallel across domains, their observations contain redundant signals that can be exploited to guide dictionary recovery toward more stable and plausible solutions. Formally,

**Definition 2** (Iso-Energy Assumption). *Let $\psi : \bigcup_{d \in \mathfrak{D}} \mathcal{X}^{(d)} \to \mathbb{R}^K$ denote the learned encoder (VLM $f$ composed with SAE $\phi$). We say $\psi$ satisfies Iso-Energy if the second moment of each coordinate is domain-invariant:*

$$\mathbb{E}_{X \in \mathcal{X}^{(d)}} \left( \psi(X)_k^2 \right) = \mathbb{E}_{X \in \mathcal{X}^{(d')}} \left( \psi(X)_k^2 \right), \quad \text{with } k \in [\![1, K]\!], \ (d, d') \in \mathfrak{D}^2.$$

Iso-Energy provides a testable, domain-agnostic inductive bias: *cross-modal* features should correspond to *bimodal* concepts and maintain comparable energy across domains, whereas *modality-specific* factors need not. This constraint narrows the solution set without requiring instance-level matching. It is reminiscent of the rosetta neurons (Dravid et al., 2023; Gresele et al., 2020) and the platonic representation hypothesis (Huh et al., 2024), suggesting that independent models (in our case, the vision and language encoders) converge on shared features.

**Operationalization.** To operationalize this assumption, we adopt as our base the Matching Pursuit (MP) sparse autoencoder, recently introduced by Costa et al. (2025) and rooted in the original framework of Mallat & Zhang (1993). MP enforces $\ell_0$ sparsity through sequential residual updates, which

---

for any $c$ there exists a neighborhood $U(c)$ on which $g^{(d)}$ is injective with continuous inverse.

aligns with the sparse generative model introduced above and has shown strong empirical reconstruction performance in vision compared to ReLU (Bricken et al., 2023), JumpReLU (Rajamanoharan et al., 2024), or BatchTopK SAEs (Gao et al., 2025; Bussmann et al., 2024) (see Appendix A for their formal definition). We then incorporate Iso-Energy into the sparse autoencoder recipe as a soft regularizer: a small penalty encouraging the activations of the same atom to maintain similar strength across domains. Given $\ell_2$-normalized codes $\boldsymbol{Z}^{(\mathrm{d})}, \boldsymbol{Z}^{(\mathrm{d}')} \in \mathbb{R}^{b \times K}$, our training loss becomes:

$$\mathcal{L}_{\mathrm{SAE-A}} = \mathcal{L}_{\mathrm{SAE}} + \beta \cdot \mathcal{L}_{\mathrm{align}} \quad \text{with} \quad \mathcal{L}_{\mathrm{align}} = -\frac{1}{b} \operatorname{Tr}\left( \boldsymbol{Z}^{(\mathrm{d})} \, \boldsymbol{Z}^{(\mathrm{d}')\top} \right) \tag{1}$$

with $\mathcal{L}_{\mathrm{SAE}}$ being the training loss of a standard SAE, typically an $\ell_2$ norm measuring reconstruction error with a sparsity constraint on the encoder's output. $\mathcal{L}_{\mathrm{align}}$ is the soft inductive bias, where $b$ is the batch size and $K$ the number of latent atoms. With a small weight ($\beta \approx 10^{-4}$, see Appendix B), this regularizer gently biases the dictionary toward *bimodal* features while preserving reconstruction performance. In Section 4, we demonstrate that this bias does not force the model to create multimodal concepts that don't exist in the input data. It is important to note that the minimum of this loss function is consistent with the iso-energy principle for cross-modal concepts formalized in Definition 2: maximizing the cosine similarity of codes coming from the aligned samples from the different modalities leads to codes with the same energy across modalities. From this point onward, we denote the unregularized model as SAE, and its alignment-augmented counterpart as SAE-A.

## 4 RECOVERING MULTIMODAL STRUCTURE WITH ALIGNED SAES

**Sanity check.** Before turning to large-scale embeddings, we validate our approach on controlled toy data with known ground truth. We construct two synthetic data-generating processes that mimic CLIP-like cosine similarity statistics (Fig. 9) while giving us exact control over which atoms are *unimodal* and which are *bimodal* (Appendix C.1). Each sample is generated by drawing a sparse code $\boldsymbol{z}$ with $\|\boldsymbol{z}\|_0 = L = 20$ and producing normalized embeddings

$$\boldsymbol{x}^{(\mu)} = \boldsymbol{z}^\top \boldsymbol{D}^{(\mu)}, \qquad \boldsymbol{x}^{(\nu)} = \boldsymbol{z}^\top \boldsymbol{D}^{(\nu)}, \qquad \|\boldsymbol{x}^{(\mu)}\|_2 = \|\boldsymbol{x}^{(\nu)}\|_2 = 1.$$

Two parameters govern the process. First, $\tau_1$ sets the cross-modal alignment of each shared (*bimodal*) atom $k \in B$ by fixing the cosine between its per-modality components. Second, $\tau_2$ fixes the average paired image–text similarity at the embedding level:

$$\cos\angle\big(\boldsymbol{d}_k^{(\mu)}, \boldsymbol{d}_k^{(\nu)}\big) = \frac{\langle \boldsymbol{d}_k^{(\mu)}, \boldsymbol{d}_k^{(\nu)} \rangle}{\|\boldsymbol{d}_k^{(\mu)}\|_2 \, \|\boldsymbol{d}_k^{(\nu)}\|_2} = \tau_1, \quad \text{and} \quad \mathbb{E}\Big[\big\langle \boldsymbol{x}^{(\mu)}, \boldsymbol{x}^{(\nu)} \big\rangle\Big] = \tau_2.$$

with the Iso-Energy case corresponding to $\tau_1 = 1$ (the bimodal atom is identical across modalities up to scale).

Using this approach, we can generate a distribution that matches the alignment statistics found in CLIP embeddings. We then compare a standard Sparse Autoencoder (SAE) to its "aligned" variant trained with the loss function in Eq. 1. This new loss encourages energy consistency without compromising reconstruction quality, as indicated by the high R-squared value ($R^2 \geq 0.99$) we observed in all our experiments. Recovery is evaluated against $(\boldsymbol{D}^\star, \boldsymbol{Z}^\star)$ using the Wasserstein distance $\mathcal{W}$ between learned and true atoms (lower is better) and mean matching accuracy (mma) of usage patterns after optimal bipartite matching (higher is better). When Iso-Energy is violated ($\tau_1 \neq 1$), both SAE and SAE-A recover the dictionary equally well ($\mathcal{W} \approx 0.19$, mma $\approx 0.82$), confirming that the regularizer does not hallucinate *bimodal* atoms. When Iso-Energy holds ($\tau_1 = 1$), the standard SAE fails ($\mathcal{W} = 0.396$, mma $= 0.29$) while the aligned SAE-A succeeds ($\mathcal{W} = 0.184$, mma $= 0.52$), showing that the inductive bias is neutral when unnecessary and decisive when appropriate. Full experimental details are provided in Appendix C.2. Having validated the principle in controlled settings, we next study its behavior on embeddings from foundation-scale VLMs.

### 4.1 EMPIRICAL EVALUATION ON VISION–LANGUAGE FOUNDATIONS

**Setup.** We train both SAE and its aligned counterpart SAE-A on activations from six representative models: CLIP (ViT-B/32, ViT-L/14) (Radford et al., 2021), OpenCLIP, OpenCLIP-L (Cherti et al., 2023), SigLIP (Zhai et al., 2023), and SigLIP2 (Tschannen et al., 2025) (see Appendix D for more details on these models). All SAE models are trained with identical hyperparameters (expansion ratio

| Metric | CLIP | | CLIP-L | | OpenCLIP | | OpenCLIP-L | | SigLIP | | SigLIP2 | |
|---|---|---|---|---|---|---|---|---|---|---|---|---|
| MSE ($\downarrow$) | **0.141** | 0.163 | **0.207** | 0.213 | **0.246** | 0.257 | **0.244** | 0.253 | **0.212** | 0.214 | **0.115** | 0.115 |
| $R^2$ ($\uparrow$) | **0.859** | 0.837 | **0.793** | 0.787 | **0.754** | 0.742 | **0.755** | 0.747 | **0.788** | 0.784 | 0.884 | **0.885** |
| $p_{\mathrm{acc}}$ ($\uparrow$) | 0.847 | **0.915** | 0.843 | **0.868** | 0.849 | **0.880** | 0.845 | **0.873** | 0.897 | **0.899** | 0.886 | **0.903** |
| $\rho$ ($\uparrow$) | 0.327 | **4.232** | 1.566 | **4.086** | 4.072 | **16.02** | 8.737 | **16.58** | 1.370 | **2.182** | 0.713 | **1.475** |
| FDA ($\uparrow$) | 2.630 | **4.559** | 3.914 | **4.800** | 4.369 | **8.160** | 9.787 | **16.49** | 8.831 | **34.95** | 8.246 | **18.24** |
| $\delta_{\mathrm{r}}$ ($\downarrow$) | 0.224 | **0.125** | 0.039 | **0.021** | 0.037 | **0.018** | 0.001 | **-0.000** | 0.023 | **0.006** | 0.007 | **-0.006** |

Table 1: **Comparison of unregularized vs. aligned sparse autoencoders across six VLMs.** We report reconstruction fidelity (MSE, $R^2$) and multimodality-sensitive metrics: probing accuracy ($p_{\mathrm{acc}}$), functional alignment ($\rho$), Functional and Distributional Agreement (FDA), and interventional robustness ($\delta_{\mathrm{r}}$). Left values correspond to SAE, right values to SAE-A. While reconstruction is nearly identical, the aligned variant consistently improves on all multimodality metrics : $\rho$ increases by more than an order of magnitude, FDA doubles (or triples), and $\delta_{\mathrm{r}}$ remains near zero, demonstrating that bimodal atoms alone sustain cross-modal alignment while unimodal ones contribute little beyond modality specific bias.

8, target $\ell_0 = 20$) using a subset of 1 million LAION embeddings chosen at random, ensuring that observed differences are attributable to the Iso-Energy regularizer rather than training artifacts. Classical reconstruction metrics (MSE, $R^2$) reveal little difference between the two methods, confirming that the alignment penalty does not compromise fidelity. Yet reconstruction alone is uninformative: two dictionaries with identical $R^2$ may encode radically different concept structures. To capture these distinctions, we introduce a suite of multimodality-sensitive metrics, each designed to probe a complementary aspect of the recovered dictionary. For the full definitions of these metrics, we refer the reader to Appendix E.

**Tuning $\beta$.** In Equation (1), we select $\beta$ via a log-sweep over $\{10^{-6}, \ldots, 10^{-1}\}$ and pick the largest value such that the difference in explained variance compared to non regularised SAE is less than 0.05. This rule reproduces our settings without hand-tuning and, under this criterion, we also observe no degenerate (always-on) features in practice; see Appendix B for details.

**Metrics:**

*(i) Probing accuracy $p_{\mathrm{acc}}$ (Appendix E.2.2).* This metric tests whether the geometry of dictionary atoms reflects the modality structure of the embedding space. *Unimodal* atoms should act as strong linear classifiers for domain membership, while *bimodal* atoms should remain domain-agnostic. $p_{\mathrm{acc}}$ gathers all these classifiers' performance in a single scalar. High $p_{\mathrm{acc}}$ therefore indicates that the dictionary correctly distinguishes modality-specific and shared information.

*(ii) Functional alignment $\rho$ (Appendix E.2.2).* Beyond geometry, we ask which features actually drive cross-modal alignment. $\rho$ measures the ratio of alignment explained by *bimodal* versus *unimodal* features through instance-level co-activation patterns. Values of $\rho > 1$ indicate that alignment is predominantly supported by *bimodal* concepts, consistent with the Iso-Energy Assumption.

*(iii) Functional and Distributional Agreement (FDA) (Appendix E.2.2).* At the population level, FDA measures whether the functional role of features is consistent with their geometric organization. While $\rho$ is local and instance-based, FDA checks distributional alignment across large batches. Intuitively, high FDA confirms that *bimodal* atoms globally sustain the match between modalities. Neither $\rho$ nor FDA considers the contrastive aspect of alignment, which is done by our final metric.

*(iv) Interventional robustness $\delta_{\mathrm{r}}$ (Appendix E.2.2).* To test for causality, we measure how performance in retrieval tasks changes when we remove *unimodal* features. We use $\delta_{\mathrm{r}}$ to quantify the change in recall after ablating these *unimodal* atoms. A small $\delta_{\mathrm{r}}$ suggests that these atoms aren't essential, while a large drop indicates they are necessary for the contrastive aspect of the embeddings.

**Results.** As shown in Table 1, SAE-A matches SAE's MSE and $R^2$ scores but consistently outperforms it on all multimodality-sensitive metrics. In particular, $\rho$ increases by more than an order of magnitude across models, indicating that alignment is almost entirely driven by *bimodal* atoms. Likewise, FDA improves substantially, confirming consistency at the distributional level. Probing accuracy improves modestly, reflecting a more distinct geometric separation, while $\delta_{\mathrm{r}}$ remains small, demonstrating that *unimodal* atoms can be safely ablated without harming retrieval performance and therefore do not contain cross-modal information. Taken together, these results offer converging evidence that, in SAE-A, *unimodal* atoms encode modality-specific information, while *bimodal*

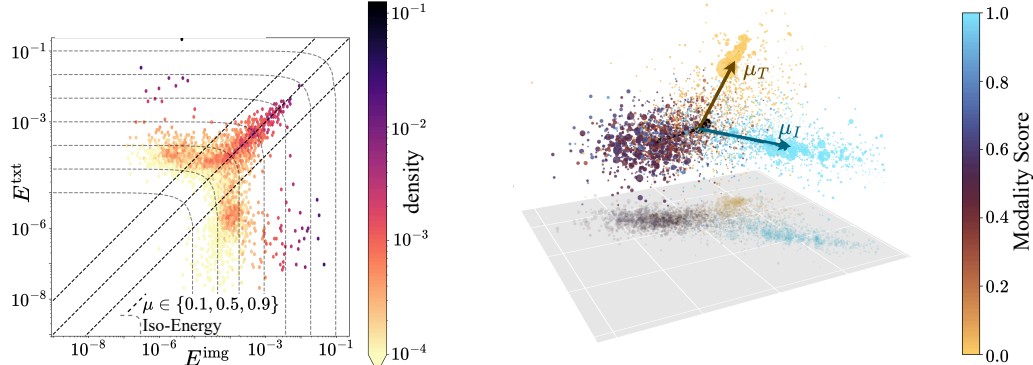

Figure 2: **(Left) Energy distribution across learned atoms.** The majority of features are *bimodal* and medium-energy (inside diagonals defined by constant modality score $\mu$-Appendix E.2.2), while only a handful of high-energy *unimodal* features dominate modality-specific variance. These high-energy *unimodal* atoms behave like modality biases and are responsible for much of the observed modality gap. **(Right) Geometric organization of concepts.** Low-dimensional projections reveal three distinct clusters: image-only, text-only, and *bimodal*. *Unimodal* atoms align with the modality cones of the embedding space, while *bimodal* atoms occupy a modality-agnostic subspace orthogonal to these directions, thereby sustaining cross-modal alignment.

atoms, identified via Iso-Energy, constitute the principal basis for cross-modal alignment. In SAE, however, the picture is less clear, and some *unimodal* atoms carry *cross-modal* information.

These findings indicate that the Iso-Energy Assumption reveals a qualitatively distinct structure: a compact *cross-modal* subspace, entirely spanned by *bimodal* concepts, that retains the contrastive power of the original embeddings while fully supporting cross-modal alignment. As we will see in Section 5, making this structure explicit also makes it actionable: it allows us to manipulate representations directly, from closing the modality gap to performing in-distribution semantic arithmetic. But before making this structure actionable, we propose an in-depth analysis of the solution to understand some aspects of the geometry.

### 4.2 CONCEPT GEOMETRY UNDER ISO-ENERGY

Having established the effectiveness of Iso-Energy on real VLMs, we now turn to a qualitative characterization of the learned concepts. This analysis focuses on how energy is distributed, how atoms are geometrically organized, and how interpretable they are in practice.

**Energy distribution.** A first observation concerns how energy is distributed across modalities. As illustrated in Figure 2, the vast majority of features are *bimodal* and exhibit moderate energy levels, whereas a small subset of *unimodal* features concentrates disproportionately high energy. These high-energy *unimodal* atoms dominate modality-specific variance and act as biases, as further detailed in Appendix F.

**Geometric organization.** The high values of $p_{acc}$ show a near-perfect alignment between latent structure and concept organization. This can be visualized by projecting the learned atoms into a low-dimensional space, revealing a clear separation into three clusters: image-only, text-only, and *bimodal* (Fig. 2, right). *Unimodal* atoms align tightly with the cones spanned by image and text embeddings, reproducing the geometry of the modality gap. In contrast, *bimodal* atoms occupy a modality-agnostic subspace, orthogonal to the *unimodal* directions. This geometry explains why *bimodal* (resp. *unimodal*) atoms carry cross-modal (resp. modality-specific) information.

**Qualitative inspection.** Finally, we examine the semantic meaning of individual atoms by inspecting their most activating examples (Appendix G). *Bimodal* atoms are semantically stable, consistently capturing the same concept across modalities (e.g., colors, objects, actions). *Unimodal* atoms, on the other hand, often reflect idiosyncratic modality-specific signals (such as poor cropping artifacts in images or "name patterns" in text) that contribute little to cross-modal semantics. Together, these

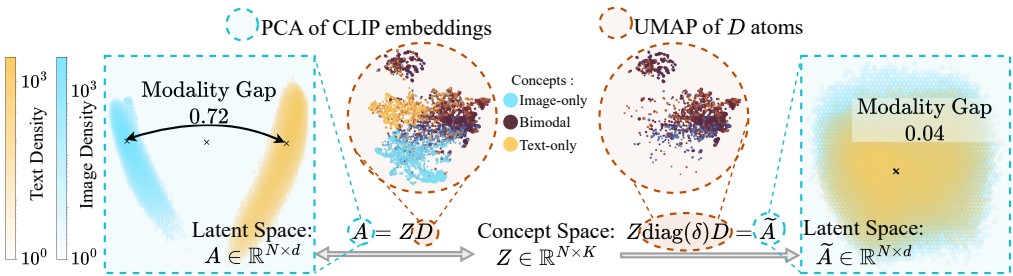

Figure 3: **The modality gap arises from multiple *unimodal* concepts, while *bimodal* concepts are sufficient to sustain cross-modal alignment. Left:** CLIP embeddings are re-expressed through a learned dictionary. A PCA projection highlights the separation between modalities, and a UMAP layout distinguishes two types of atoms: *unimodal* and *bimodal*. **Right:** Removing *unimodal* atoms with a binary mask $\boldsymbol{\delta} \in \{0,1\}^K$ closes the gap. The reconstructed embeddings $\widetilde{A}$ continue to support retrieval, indicating that *bimodal* atoms alone capture the structure necessary for alignment.

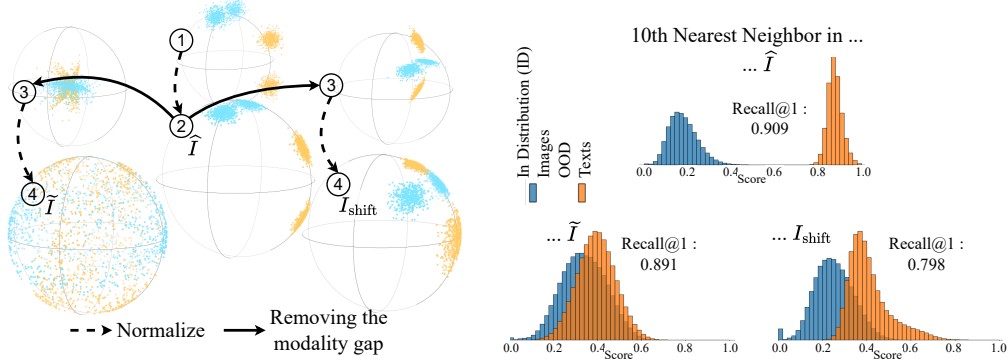

Figure 4: **Filtering *unimodal* atoms closes the modality gap without harming performance. (Left)** Synthetic illustration comparing our method with the embedding shift baseline (Liang et al., 2022). Only our approach merges image and text distributions. **(Right)** Histogram of distances from each image (ID) and caption (OOD) embedding to its 10th nearest image neighbor. The modality gap is measured as the separation between the ID and OOD histograms. Filtering *unimodal* atoms aligns the two distributions, whereas shift degrades performance and leaves the gap wide open.

three perspectives converge on the same conclusion: ***unimodal* atoms function as modality-specific biases, while *bimodal* atoms encode the shared conceptual backbone that supports cross-modal alignment.**

## 5  ACTIONABLE INTERVENTIONS ON MULTIMODAL EMBEDDINGS

Together, these analyses demonstrate that our dictionaries yield a structured and interpretable decomposition of multimodal embeddings. Building on this foundation, we now shift from analysis to intervention: once Iso-Energy isolates the cross-modal backbone, it enables direct manipulation of embeddings in ways that were previously inaccessible. In particular, we consider the minimal intervention that removes modality information and examine its effect on two structural aspects. In fact, we show that such transformations are possible without altering ranking-related capabilities even when the modality information is non-trivial (e.g., not a bias, as proposed by Zhang et al. (2023)'s Proposition 1) and under realistic assumptions of orthogonality.

**Proposition 1** (Modality information removal impact on ranking.)**.** *Consider $\boldsymbol{v} \in \mathbb{R}^d$ with decomposition $\boldsymbol{v} = \omega(\boldsymbol{x}) + \gamma(\boldsymbol{x})$ where $\omega(\boldsymbol{x}) \in \Omega$ encodes modality-specific information, $\gamma(\boldsymbol{x}) \in \Gamma$ captures cross-modal content, and $\mathbb{R}^d = \Omega \oplus \Gamma$. If visual and textual information are orthogonal, then ranking preservation is guaranteed.*

*Proof.* See Appendix K. □

**Closing the modality gap.**    First, we find that by filtering out *unimodal* atoms with a binary mask (Figure 3), we nearly eliminate the modality gap while preserving retrieval and zero-shot performance.

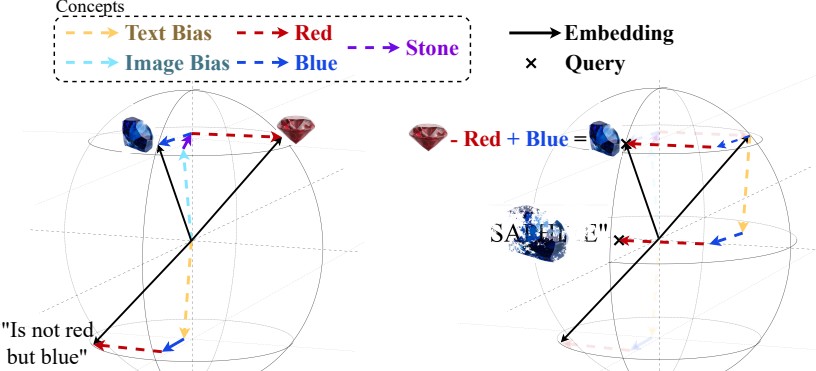

Figure 5: **Semantic vector arithmetic restricted to cross-modal information.** Starting from a Ruby (red stone), the target is a Sapphire (blue stone). The classical edit vector $\Delta = \text{Text} + \text{Blue} - \text{Red}$ is polluted by *unimodal* directions, producing a query $Q = I_{\text{src}} + \Delta$ that drifts out-of-distribution. In contrast, restricting to *bimodal* atoms yields $Q_{\text{SAE}}$, which lies on the semantic manifold and reliably retrieves the correct target. This illustrates how *unimodal* features inject modality-specific bias into $\Delta$, while Iso-Energy isolates the truly shared concepts that support valid semantic arithmetic.

As illustrated in Figure 4, this intervention merges the image and text distributions, unlike embedding shift baselines (Liang et al., 2022), which enforce matching means but still leave distributions well separated. Crucially, our approach preserves contrastive capabilities, showing that our dictionaries faithfully capture both contrastive and modality-specific information separately through *bimodal* and *unimodal* atoms. Measuring the modality gap is typically done by measuring the distance between the mean of text and image distributions, or by measuring their linear separability. However, these fail to account for distributional mismatches remaining post intervention. For this reason, we chose to turn to the out-of-distribution (OOD) literature to measure the modality gap, and borrow a method described by Sun et al. (2022). This method measures the separation between the blue and orange histograms in Figure 4. Our method consists in the following intervention on the activations, indicated with a tilde: $\widetilde{A} := (Z \odot \delta)D$, where $Z \in \mathbb{R}^{N \times K}$ contains the sparse codes, $D$ is the concept dictionary, and $\delta$ is the binary mask filtering out *unimodal* features, broadcast to the size of $Z$. Reconstructed activations are indicated with a hat: $\widehat{A} := ZD$. The embedding shift intervention, and variants described in the appendix, consists in adding a modality-wise constant, essentially moving the mean of each distribution $\mu_{I/T}$. Liang et al. (2022)'s shift method transforms the images by $I_{\text{shift}} := I - \mu_I + \frac{\mu_I + \mu_T}{2}$, similarly for texts $T_{\text{shift}}$.

**Semantic vector arithmetic.** Iso-Energy also grounds semantic manipulations. Let $I_{\text{src}}$ be the source image, and $\Delta$ be the textual description of the difference between the source and target image – i.e., the relative caption. Restricting vector arithmetic to *bimodal* atoms $Q_{\text{SAE}} := I_{\text{src}} + \widetilde{\Delta}$ (Figure 5) produces queries that remain in-distribution (Figure 6 and Table 2) while preserving retrieval (Appendix H). In contrast, classical arithmetic $Q = I_{\text{src}} + \Delta$ incorporates *unimodal* noise from the text embedding on top of the interesting cross-modal information, yielding degenerate queries that drift outside the embedding distribution. Our intervention consistently produces queries in-distribution without degrading performance on the FashionIQ benchmark Wu et al. (2021), demonstrating that the *bimodal* backbone revealed by Iso-Energy is practically useful.

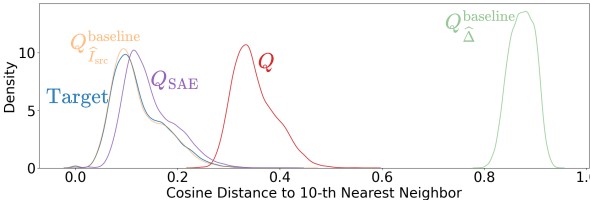

Figure 6: **Out-of-distribution behavior of semantic queries.** Histogram of distances between each query and its 10th nearest neighbor in the target image distribution. Classical arithmetic $Q = I_{\text{src}} + \Delta$ drifts out-of-distribution, while our *bimodal*-restricted query $Q_{\text{SAE}}$ remains aligned with the target space. Baselines using only the source image ($Q_{\widehat{I}_{\text{src}}}^{\text{baseline}}$) or only the caption difference ($Q_{\widehat{\Delta}}^{\text{baseline}}$) confirm the expected extremes: perfectly in-distribution and fully OOD, respectively.

| OOD score (↓) | $Q$ | $Q_{\text{SAE}}$ |
|---|---|---|
| CLIP | 0.97 | **0.77** |
| CLIP-L | 0.95 | **0.76** |
| OpenCLIP | 0.86 | **0.68** |
| OpenCLIP-L | 0.87 | **0.72** |
| SigLIP | 0.99 | **0.70** |
| SigLIP2 | 0.99 | **0.61** |

Table 2: OOD scores for classical queries vs. concept-based queries $Q_{\text{SAE}}$. Lower is better, best results per line in bold.

Consider the following example. Let $I_{\mathrm{src}}$ be an image of a ruby, $\Delta$ be the prompt *"is not red but blue"* and the target be an image of a saphire. This is the example illustrated in Figure 5. Adding $\Delta$ to the ruby produces a query that contains both textual-only and visual-only concepts and therefore that do not correspond to any realistic embedding. Adding only the cross-modal concepts of $\Delta$, however, produces a query that actually corresponds to an image of a saphire.

## 6 CONCLUSION

The Iso-Energy Assumption introduces a simple yet effective inductive bias for analyzing multimodal representations. In synthetic settings, it facilitates the recovery of ground-truth structure; in large-scale vision–language models, it consistently reveals a compact *bimodal* basis that supports cross-modal alignment. This basis makes multimodal concepts accessible, isolates *unimodal* concepts, closes the modality gap, and enables controlled semantic edits even in foundation-scale VLMs. In contrast, standard sparse autoencoders tend to learn a diffuse mixture of *unimodal* and *bimodal* atoms despite similar reconstruction quality, which obscures the structure that underpins alignment.

Despite its promising results, our approach has several limitations. First, the alignment penalty is sensitive to the choice of its weighting coefficient $\beta$: when too small, it becomes ineffective; when too large, it can lead to degenerate feature representations. In this work, we select $\beta$ through simple sweeps, but a more principled calibration—e.g., by coupling the penalty to a performance constraint or stability criterion—remains an important direction for future work. Second, our analysis is conducted on reconstructions produced by the sparse autoencoder, rather than on the original embeddings. This constrains the reported performance to the autoencoder's reconstruction regime. Finally, our experiments are limited to dual-encoder vision–language models. Whether the same structural invariants and alignment properties hold in models with cross-attention mechanisms or generative training objectives remains an open question.

More broadly, our results support a hypothesis-driven approach to interpretability: inductive biases should be selected to reflect the structural properties relevant to downstream tasks, rather than applied indiscriminately. When properly aligned with task-relevant objectives, even simple biases can illuminate underlying mechanisms that might otherwise remain hidden—and offer actionable control without compromising core performance.

## ACKNOWLEDGEMENT

Our work has benefited from the AI Cluster ANITI and the research program DEEL. ANITI is funded by the France 2030 program under the Grant agreement n°ANR-23-IACL-0002. DEEL is an integrative program of the AI Cluster ANITI, designed and operated jointly with IRT Saint Exupéry, with the financial support from its industrial and academic partners and the France 2030 program under the Grant agreement n°ANR-10-AIRT-01.

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
