# Appendix

## Table of Contents

## A SAE-DEX

In this appendix, we explore several SAE algorithms popular in the community. We provide definitions along with interpretations and implicit inductive biases for each algorithm. For similar insights into architectures, refer to Hindupur et al. (2025).

**Notations.** Let's first redefine all necessary notations here. Let $A \in \mathbb{R}^{n \times d}$ denote the collection of $n$ activations of dimension $d$ to be disentangled. Let $D \in \mathbb{R}^{K \times d}$ be the matrix containing the $K$ dictionary atoms, $Z \in \mathbb{R}^{n \times K}$ the collection of sparse codes.

Let $\phi : x \in \mathbb{R}^d \to \Pi\{Wx + b\} \in \mathbb{R}^K$ be an encoder attempting to solve the *sparse coding* optimization problem, with $W \in \mathbb{R}^{K \times d}$ and $b \in \mathbb{R}^K$ the parameters of an affine transformation on the activations, and $\Pi : \mathbb{R}^K \to \mathbb{R}^K$ a non linear projection function.

**SAE.** Recent work used SAEs to disentangle latent spaces, within the framework of the *Linear Representation Hypothesis* (LRH) (Elhage et al., 2022; Wattenberg & Viégas, 2024). This framework hypothesizes that latent space's activations arise from a sparse linear combination of semantic atoms. This linear combination should be linearly accessible, as all operations on these spaces are linear. SAEs attempt to find this combination. In practice, purely linear SAEs perform very poorly, and some non-linearity has to be introduced to the encoder to increase its expressiveness.

**ReLU.** This is the most common nonlinearity, where $\Pi = \mathrm{ReLU} : x \to \max(\mathbf{0}, \boldsymbol{x})$, making the SAE effectively a one layer MLP. Sparsity is enforced through a soft constraint, encoded in the loss typically as an $L_1$ penalty. This soft constraint results in desirable and undesirable inductive biases, like sparsity and activation shrinkage, respectively.

This projection assumes linear separability of concepts, with neurons' receptive fields being half spaces, independent of one another. This independence creates an inductive bias preventing the discovery of concepts too close from one another, as they become harder to separate.

**JumpReLU.** Rajamanoharan et al. (2024) proposed to use a gating mechanism on top of the $\mathrm{ReLU}$, as defined by Equation (2). This gate intuitively prevents noisy small activations, with concepts being active only if they are significantly present. This is not allowed by a simple $\mathrm{ReLU}$, as changing a concept's activation threshold can only happen through the bias, which also shifts its activation magnitude. Sparsity is often achieved by optimizing $\theta$ instead of using the $L_1$ penalty. This projection also assumes linear separability of concepts, with receptive fields being again independent half-spaces.

$$\Pi_\theta : x \to xH(x - \theta) \tag{2}$$

where $H$ is the Heaviside step function.

**TopK.** These SAEs were introduced by Gao et al. (2025) to facilitate training by enforcing a hard constraint on sparsity, rather than the soft constraint, as an inductive bias in the loss of the $\mathrm{ReLU}$ SAE. Indeed, TopK selects the $k$ most active concepts for each sample, enforcing an $L_0$ of at most $k$. This also prevents known, undesirable effects of the inductive bias. This projection assumes angular separability of concepts, with receptive fields being polyhedral cones such that at each point in space, at most $k$ of them overlap.

This projection introduces competition between concepts—they are no longer independent of one another—which also prevents the discovery of features too close to one another, though competition introduces different dynamics from the independent case.

$$\Pi : x \to x \odot \mathbf{1}_{\{i \in \mathcal{T}_k(x)\}}, \text{ where } \mathcal{T}_k(x) = \arg\underset{k}{\mathrm{top}}(x) \tag{3}$$

**BatchTopK.** This is a variant of the TopK algorithm, where competition is introduced in the whole batch rather than in each individual sample. With a batch size of $b$, this function selects the $b \times k$ most active concepts in the batch. This allows for a relaxed constraint on the $L_0$, which is now enforced to be $k$ on average, but allowing for sample-wise variations. This projection also assumes angular separability.

**Matching Pursuit.** As described in Algorithm 1, the Matching Pursuit algorithm iteratively adds the best atom $d_i$ to the code $z$, removes it from the residual $r$, and starts again. Two main variants exist to decide the number of iterations: one where it is fixed, and one where the algorithm continues until some threshold reconstruction is reached. We opted for the former to have better control over sparsity.

---

**Algorithm 1** Matching Pursuit SAE Encoder

---

1: **Input:** $x \in \mathbb{R}^d$, $D \in \mathbb{R}^{K \times d}$, $\lambda > 0$
2: $z \leftarrow 0 \in \mathbb{R}^K$        ▷ Initialize sparse code
3: $r \leftarrow x$        ▷ Initialize residual
4: **for** $k \in [\![1, \kappa]\!]$ **do**        ▷ Iterate for $\kappa$ steps - enforcing an $L_0$ sparsity constraint of $\kappa$
5:      $i \leftarrow \arg\max_i \left| D_i^\top r \right|$        ▷ Find the atom that best matches the residual
6:      $z_i \leftarrow D_i^\top r$        ▷ Update sparse code for this atom
7:      $r \leftarrow r - z_i D_i$        ▷ Update residual
8: **end for**

---

The main advantage of this algorithm over TopK is that each atom selected can only decrease the error term $r$. Thus, even though competition still exists, this algorithm cannot select two competing atoms

with large activations that would result in an increased error. The main downside of this algorithm is its increased complexity.

Receptive fields are again polyhedral cones, such that exactly $k$ intersect at every point in space, but both their shape and the activation patterns inside them are now much more flexible than for TopK cones. This removes the inductive bias that constrains the geometric similarity between two atoms, which is present in all other methods. It might cause a bias to learn duplicates of a feature, especially high-energy ones, to fit noise at no additional cost.

| Metric | ReLU | JumpReLU | TopK | BatchTopK | SAE | SAE-A |
|---|---|---|---|---|---|---|
| **Sparse Reconstruction** | | | | | | |
| MSE ($\downarrow$) | 0.31 | 0.26 | 0.22 | 0.22 | 0.14 | 0.16 |
| $R^2$ ($\uparrow$) | 0.69 | 0.74 | 0.78 | 0.78 | 0.86 | 0.84 |
| $\ell_0$ ($\downarrow$) | 19.8 | 17.4 | / | / | / | / |
| $\ell_1$ ($\downarrow$) | 1.22 | 2.48 | 2.59 | 2.65 | 2.94 | 3.09 |
| **Consistency** | | | | | | |
| $C-$insertion ($\uparrow$) | 0.81 | 0.79 | 0.75 | 0.76 | 0.74 | 0.71 |
| $C-$deletion ($\downarrow$) | 0.16 | 0.16 | 0.23 | 0.20 | 0.10 | 0.10 |
| Stability ($\downarrow$) | 0.067 | 0.23 | 0.31 | 0.33 | 0.15 | 0.22 |
| **Structure in D** | | | | | | |
| Stable Rank ($\uparrow$) | 88.3 | 231 | 29.6 | 161 | 28.3 | 14.0 |
| Eff. Rank ($\uparrow$) | 500 | 501 | 498 | 501 | 455 | 435 |
| Stable Rank (w) ($\uparrow$) | 1.57 | 1.46 | 1.92 | 1.32 | 1.59 | 1.17 |
| Eff. Rank (w) ($\uparrow$) | 18.0 | 37.8 | 55.7 | 46.4 | 160 | 90.3 |
| Coherence ($\downarrow$) | 0.99 | 0.50 | 0.51 | 0.55 | 0.99 | 0.99 |
| **Structure in Z** | | | | | | |
| Stable Rank ($\downarrow$) | 1.97 | 1.69 | 1.76 | 1.61 | 9.03 | 1.26 |
| Eff. Rank ($\downarrow$) | 46.0 | 45.5 | 85.4 | 80 | 829 | 532 |
| Connectivity ($\uparrow$) | 0.24 | 0.25 | 0.45 | 0.021 | 0.007 | 0.058 |
| Neg. Inter. ($\downarrow$) | 0.0008 | 0.0019 | 0.0003 | 0.0006 | 0.0005 | 0.0020 |
| **Modality** | | | | | | |
| $\rho$ ($\uparrow$) | 0.46 | 0.96 | 1.98 | 0.97 | 0.33 | 4.23 |
| $\delta_{r@1}$ ($\downarrow$) | 0.134 | 0.06 | 0.05 | 0.05 | 0.22 | 0.12 |
| FDA ($\uparrow$) | 2.52 | 3.44 | 8.77 | 3.83 | 2.63 | 4.56 |
| $p_{acc}$ ($\uparrow$) | 0.87 | 0.87 | 0.88 | 0.89 | 0.85 | 0.92 |

Table 3: Metric comparison across SAE architectures under fixed hyperparameters (expansion ratio: 8, target $\ell_0$: 20). We only report results for SAEs trained on CLIP here.

| Metric | ReLU | JumpReLU | TopK | BatchTopK | SAE | SAE-A |
|---|---|---|---|---|---|---|
| **Sparse Reconstruction** | | | | | | |
| MSE ($\downarrow$) | 0.28 | 0.19 | 0.16 | 0.16 | 0.07 | 0.08 |
| $R^2$ ($\uparrow$) | 0.72 | 0.81 | 0.84 | 0.84 | 0.93 | 0.92 |
| $\ell_0$ ($\downarrow$) | 50 | 49.1 | / | / | / | / |
| $\ell_1$ ($\downarrow$) | 1.60 | 4.38 | 3.99 | 4.43 | 4.58 | 4.69 |
| **Consistency** | | | | | | |
| $C-$insertion ($\uparrow$) | 0.75 | 0.61 | 0.61 | 0.59 | 0.59 | 0.58 |
| $C-$deletion ($\downarrow$) | 0.24 | 0.31 | 0.33 | 0.33 | 0.15 | 0.16 |
| Stability ($\downarrow$) | 0.11 | 0.35 | 0.33 | 0.36 | 0.26 | 0.32 |
| **Structure in D** | | | | | | |
| Stable Rank ($\uparrow$) | 346 | 376 | 347 | 383 | 207 | 191 |
| Eff. Rank ($\uparrow$) | 511 | 510 | 511 | 511 | 494 | 493 |
| Stable Rank (w) ($\uparrow$) | 1.65 | 1.92 | 1.85 | 2.82 | 1.60 | 1.54 |
| Eff. Rank (w) ($\uparrow$) | 26.5 | 146 | 101 | 101 | 74.8 | 55.1 |
| Coherence ($\downarrow$) | 0.59 | 0.44 | 0.51 | 0.41 | 0.99 | 0.99 |
| **Structure in Z** | | | | | | |
| Stable Rank ($\downarrow$) | 1.86 | 1.58 | 1.68 | 1.43 | 4.06 | 2.80 |
| Eff. Rank ($\downarrow$) | 48.9 | 101 | 107 | 105 | 762 | 597 |
| Connectivity ($\uparrow$) | 0.56 | 0.91 | 0.81 | 0.56 | 0.12 | 0.13 |
| Neg. Inter. ($\downarrow$) | 0.0012 | 0.0012 | 0.0017 | 0.0045 | 0.0013 | 0.0013 |
| **Modality** | | | | | | |
| $\rho$ ($\uparrow$) | 1.71 | 1.41 | 1.77 | 2.71 | 1.13 | 2.69 |
| $\delta_{\text{r@1}}$ ($\downarrow$) | -0.024 | 0.028 | 0.024 | 0.089 | 0.13 | 0.052 |
| FDA ($\uparrow$) | 8.12 | 5.17 | 5.12 | 17.3 | 2.79 | 2.57 |
| $p_{\text{acc}}$ ($\uparrow$) | 0.80 | 0.81 | 0.83 | 0.89 | 0.84 | 0.93 |

Table 4: Metric comparison across SAE architectures under fixed hyperparameters (expansion ratio: 64, target $\ell_0$: 50). We only report results for SAEs trained on CLIP here.

## B    TUNING THE STRENGTH OF $\mathcal{L}_{\text{align}}$

In this appendix, we look at the impact of the alignment loss, and how to tune its strength—which we call $\beta$ in this section. We consider the CLIP model, and train a family of SAEs with $\beta$ ranging from $10^{-5}$ to $10^{-2}$ on a logarithmic scale. The alignment loss is defined by the average cosine similarity between matching image–caption codes :

$$\mathcal{L}_{\text{align}}( \underbrace{\widetilde{Z}^{(\text{d})}}_{\in \mathbb{R}^{b \times K}}, \underbrace{\widetilde{Z}^{(\text{d}')}}_{\in \mathbb{R}^{b \times K}} ) = -\frac{1}{b}\text{Tr}(\widetilde{Z}^{(\text{d})} \cdot \widetilde{Z}^{(\text{d}')\top}) \tag{4}$$

where $b$ is the batch size, $K$ is the number of latent concept dimensions and $\widetilde{Z}^{(\text{d})}$ and $\widetilde{Z}^{(\text{d}')}$ are normalized codes.

We observe a sharp transition, with almost no effect below $\beta = 10^{-4}$, and systematic degenerate solutions above $\beta = 10^{-3}$. Figure 7 shows the evolution of reconstruction error against $\beta$. At the transition, constant features start appearing with frequencies of 1. These features are degenerate solutions to the alignment loss. Figure 8 shows the proportion of energy contained in *bimodal* and degenerate features against $\beta$.

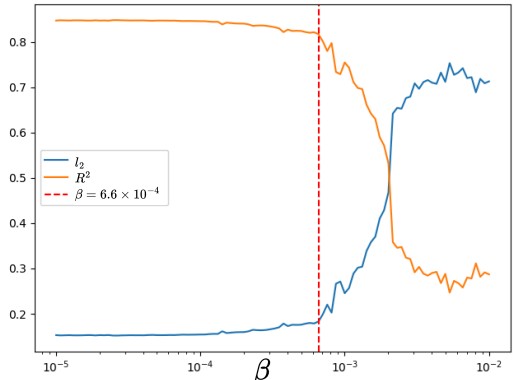

Figure 7: Reconstruction error ($MSE = l_2$) and explained variance ($R^2$) against the strength of the alignment penalty ($\beta$).

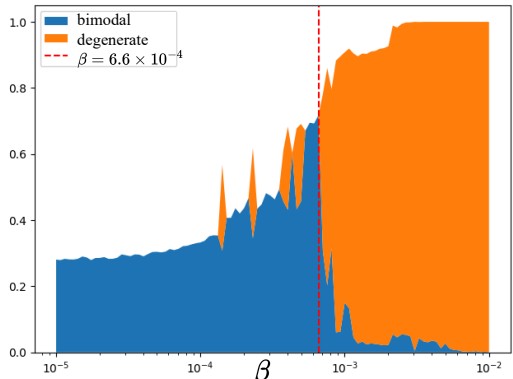

Figure 8: Proportion of energy contained in *bimodal* features, and in degenerate features, against the strength of the alignment penalty.

## C   TOY DATA GENERATING PROCESS

### C.1   DGP DEFINITION

For our toy Data Generating Process (DGP), we created two ground truth dictionaries $D_1, D_2$ and sparse codes $Z_1, Z_2$ such that $Z_1 D_1 = Z_2 D_2$. The goal is to have a realistic-enough distribution on which we have full control of which features are "*unimodal*" and which are "*bimodal*". $(D_1, Z_1)$ will have "ground truth" *bimodal* features while $(D_2, Z_2)$ will contain "ground truth" *unimodal* features. One parameter, $\tau_1$ controls smoothly the separation between these, with no ambiguity over which of the two is the "true" ground truth when this parameter is extreme. When $\tau_1$ is close to 1—resp. $\tau_2$—$(D_1, Z_1)$—resp. $(D_2, Z_2)$— is the ground truth dictionary, without ambiguity. Below is the list of parameters :

- $d$, the dimension of the latent space.
- $k$, controlling the number of concepts, such that $(D_1, Z_1)$ has $10k$ concepts and $(D_2, Z_2)$ has $14k$ concepts.
- $L$ : the sparsity of the codes, such that $\forall i, L_0(Z_{2,i}) = L$.
- $\tau_1$ : controls the cosine similarity between $D_2^I$ and $D_2^T$, such that $\forall i, D_{2,i}^{I,\top} \cdot D_{2,i}^T = \tau_1$
- $\tau_2$ : controls the cosine similarity between a matching image-caption pair, such that $\forall i, (Z_{2,i}^I D_2)^\top (Z_{2,i}^T D_2) = \tau_2$

where the superscript $\cdot^I$ indicates that we take the submatrices corresponding to image codes in $Z$ or to image-only features in $D$. Similarly for $\cdot^T$ and text.

$(\boldsymbol{D_1, Z_1})$. Let $D^I$ and $D^T$ be subsets of concepts of size $k$ each and corresponding to high-energy high-frequency *unimodal* concepts. Let $D^B$ be $4k$ *bimodal* concepts, associated to small (depending on $\tau_1$) additional terms $D^{B,I}$ and $D^{B,T}$ of size $4k$ each. These additional terms are modality-specific and control for the fact that features can have some amount of *unimodal* information.

We enforce that all features in $D^I$ are orthogonal to all features in $D^T$, $D^B$ and $D^{B,T}$, similarly for $D^{B,I}$, $D^T$ and $D^{B,T}$. All features in $D^B$ are thus orthogonal to all features not in $D^B$.

To generate one sample $Z_{1,i}$, we proceed as follows. We select exactly one feature in $D^I$ and one in $D^T$ that will have an activation of 1 (temporarily). Then, we select $L-1$ features in $D^B$, and their corresponding additional terms in $D^{B,I}$ and $D^{B,T}$. The activations associated to $D^B$ are constant, as are those in $D^{B,I}$ and $D^{B,T}$, and equal to $\tau_1 \beta$ and $(1-\tau_1)\beta$ respectively. To compute $\beta$, we select the only positive real-valued root of the underlying 4th-degree polynomial to satisfy the constraint given by $\tau_2$. Finally, we scale all activations by $\frac{1}{\|Z_{1,i}D_1\|}$ such that the final embeddings have unit norm.

$(\boldsymbol{D_2, Z_2})$. $D_2$ is the collection of features in $D^I$, $D^T$ as well as $\tau_1 D^B + (1-\tau_1)D^{B,I}$ and $\tau_1 D^B + (1-\tau_1)D^{B,T}$. Activation strengths in $Z_2$ associated to $B^I$ and $B^T$ are unchanged, and those associated to $\tau_1 D^B + (1-\tau_1)D^{B,I/T}$ are equal to $\beta$.

In Appendix C.2, we use $\tau_2$ to have cosine similarity histograms similar to those obtained on CLIP embeddings of LAION, and for $\tau_1$, we use $\tau_2 + 0.1$ in experiment 1, and $0.999$ in experiment 2.

In Figure 9 we show the histogram of cosine similarities between embeddings generated with CLIP and LAION, as well as those between embeddings generated by our DGP.

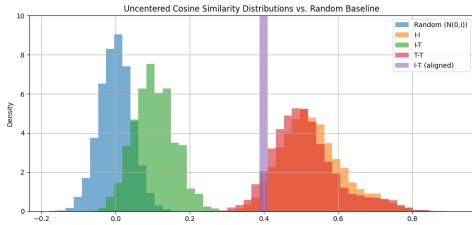 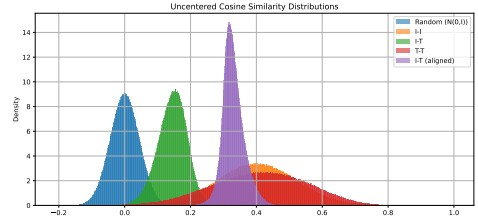

Figure 9: Histograms of cosine similarities between CLIP embeddings (**right**). "II": image-image embedding pairs, "TT": text-text, "IT": random image-caption, "IT (aligned)": matching image caption, Random: here for reference. **Left:** cosine similarity histogram for the embeddings generated by our toy DGP.

## C.2 EXPERIMENTS

In this section, we investigate the impact of enforcing our alignment penalty $\mathcal{L}_{\text{align}}$ on the learned dictionaries. For that, we design two toy datasets sampled from two different data-generating processes (DGPs), one satisfying the Iso-Energy Assumption and one violating it. In both cases, parameters of the DGPs are chosen such that the data distribution is similar to the actual data distribution of LAION-400M embedded by CLIP ViT-B/32. Details on the DGPs and the datasets are given in Appendix C.1.

We train our SAEs using the same number of atoms and sparsity as those of the DGPs. We do not report sparse reconstruction metrics, as they are all equally good, with $R^2 \geq 0.99$ and fixed sparsity.

**Experiment 1.** In our first experiment, we show that both SAE and SAE-A are equally able to find the ground truth dictionary in a setup where the Iso-Energy Assumption is violated. In other words, we check that enforcing the alignment penalty does not lead to learning spurious *bimodal* features.

Let $D$ and $Z$ be the learned dictionary and sparse codes, respectively. Let $W$ and $C$ be the ground truth dictionary weights and corresponding sparse codes, respectively. We measure the proximity

between both dictionaries using the Wasserstein distance $\mathcal{W}(D, W)$ between the ground truth and the learned dictionary's atoms. We also measure the mean matching accuracy $\mathrm{mma}(Z, C)$, for a permutation matrix $\Pi \in \mathbb{R}^{K \times K}$ that best matches the atoms in $Z$ and $C$. $\mathrm{mma}(Z, C)$ is the average cosine similarity between matched atoms' activation patterns in $Z$ and $C$.

Results are summarized in Table 5a. We observe no distinction between the two SAEs. In particular, we show that adding the alignment penalty $\mathcal{L}_{\mathrm{align}}$ does not incentivize SAE-A to learn spurious *bimodal* features.

**Experiment 2.** In our second experiment, the DGP from which the data is sampled satisfies the Iso-Energy Assumption. In this case, we observe that training a SAE fails to uncover the ground truth dictionary and learns spurious, *unimodal* features, while training a SAE-A succeeds at doing so. Results are summarized in Table 5b.

| | $\mathcal{W}(D, W) \downarrow$ | $\mathrm{mma}(Z, C) \uparrow$ |
|---|---|---|
| SAE | 0.197 | 0.83 |
| SAE-A | 0.185 | 0.81 |

(a) Both SAE and SAE-A learn an equally good approximation of the ground truth dictionary. **Adding the alignment penalty $\mathcal{L}_{\mathrm{align}}$ does not lead to learning spurious *bimodal* features**.

| | $\mathcal{W}(D, W) \downarrow$ | $\mathrm{mma}(Z, C) \uparrow$ |
|---|---|---|
| SAE | 0.396 | 0.29 |
| SAE-A | **0.184** | **0.52** |

(b) **SAE fail to learn ground truth *bimodal* atoms**. SAE-A learns the ground truth dictionary.

Table 5: Comparision between SAE and SAE-A (trained with $\mathcal{L}_{\mathrm{align}}$). $\mathcal{W}(D, W)$ : Wasserstein distance between learned and ground truth dictionaries (lower is better). $\mathrm{mma}(Z, C)$ : mean matching accuracy between learned and ground truth sparse codes (higher is better). **(a)** Experiment 1: the data-generating process violates the Iso-Energy Assumption. **(b)** Experiment 2: the data-generating process satisfies the Iso-Energy Assumption.

## D  MODEL-DEX

We studied a collection of models, varying in size and training procedure from the CLIP (Radford et al., 2021), SigLIP (Zhai et al., 2023), and OpenCLIP (Cherti et al., 2023) implementations.

**CLIP and OpenCLIP.** We used the ViT-B/32 and ViT-L/14 variants of both CLIP and OpenCLIP. These were trained with the contrastive softmax loss to align matching pairs and separate non-matching ones. Specifically, we used `openai/clip-vit-base-patch32`, `openai/clip-vit-large-patch14`, `laion/CLIP-ViT-B-32-laion2B-s34B-b79K` and `laion/CLIP-ViT-L-14-laion2B-s32B-b82K` from hugging face's `transformers.CLIPModel` implementation. They are referred to as CLIP, CLIP-L, OpenCLIP, and OpenCLIP-L.

**SigLIP.** We conducted our experiments using both SigLIP, trained with a sigmoid-based contrastive loss, and SigLIP2, trained with a heterogeneous mix of objectives. Specifically, we used `google/siglip-base-patch16-224` and `google/siglip2-base-patch16-224` from hugging face's `transformers.SiglipModel` implementation.

We report in Figure 10 linear projections of embeddings, colored by modality, illustrating the modality gap. We measure the modality gap both as the Difference in Mean (DiM) between each modality and as the difference in distribution, with the Wasserstein distance ($\mathcal{W}$). We repeat this on three datasets (Appendix J). We report the modality gap inside of each dataset in Table 6.

Finally, we report recall@1 and recall@5, along with histograms of cosine similarities between embeddings of the same modality, matching embeddings of different modalities, and random pairs of embeddings of different modalities in Figure 11 and Table 6. We add a histogram of cosine similarity between samples from white noise as a reference.

For the ImageNet dataset, we also measure zero-shot recall—slightly different from recall@1, see Appendix E—and classifier accuracy.

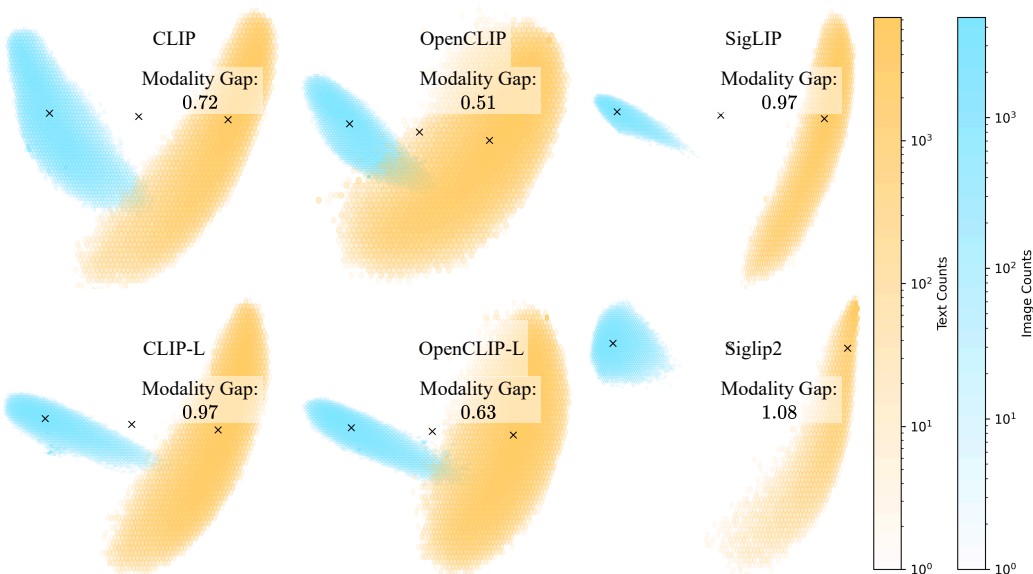

Figure 10: PCA of image (blue) and text (yellow) embeddings from LAION for different VLM encoders. The modality gap is measured as the difference in mean (DiM) and as the Wasserstein distance ($\mathcal{W}$) between the two distributions. A cross indicates the modality-wise and global means.

| | CLIP | CLIP-L | OpenCLIP | OpenCLIP-L | SigLIP | SigLIP2 |
|---|---|---|---|---|---|---|
| **LAION** | | | | | | |
| DiM $\mid \mathcal{W}$ | 0.72 $\mid$ 0.68 | 0.70 $\mid$ 0.73 | 0.51 $\mid$ 0.70 | 0.63 $\mid$ 0.75 | 0.97 $\mid$ 0.94 | 1.08 $\mid$ 0.90 |
| recall@1 $\mid$ @5 | 0.97 $\mid$ 0.99 | 0.97 $\mid$ 0.99 | 0.98 $\mid$ 0.99 | 0.98 $\mid$ 0.99 | 0.96 $\mid$ 0.99 | 0.36 $\mid$ 0.46 |
| **COCO** | | | | | | |
| DiM $\mid \mathcal{W}$ | 0.82 $\mid$ 0.67 | 0.81 $\mid$ 0.72 | 0.72 $\mid$ 0.65 | 0.75 $\mid$ 0.68 | 1.04 $\mid$ 0.90 | 1.05 $\mid$ 0.86 |
| recall@1 $\mid$ @5 | 0.71 $\mid$ 0.95 | 0.74 $\mid$ 0.96 | 0.77 $\mid$ 0.97 | 0.81 $\mid$ 0.98 | 0.81 $\mid$ 0.98 | 0.81 $\mid$ 0.97 |
| **ImageNet** | | | | | | |
| DiM $\mid \mathcal{W}$ | 0.86 $\mid$ 0.68 | 0.86 $\mid$ 0.73 | 0.77 $\mid$ 0.67 | 0.84 $\mid$ 0.69 | 1.13 $\mid$ 0.93 | 1.13 $\mid$ 0.89 |
| recall@1 $\mid$ @5 | 0.65 $\mid$ 0.91 | 0.74 $\mid$ 0.95 | 0.67 $\mid$ 0.92 | 0.73 $\mid$ 0.95 | 0.75 $\mid$ 0.96 | 0.76 $\mid$ 0.97 |
| zero-shot accuracy | 0.60 $\mid$ 0.85 | 0.73 $\mid$ 0.92 | 0.63 $\mid$ 0.86 | 0.72 $\mid$ 0.91 | 0.73 $\mid$ 0.92 | 0.75 $\mid$ 0.93 |
| Classifier acc | 0.75 $\mid$ 0.94 | 0.84 $\mid$ 0.97 | 0.77 $\mid$ 0.94 | 0.83 $\mid$ 0.97 | 0.83 $\mid$ 0.97 | 0.84 $\mid$ 0.97 |

Table 6: Modality Gap and recall on three datasets for the six models studied. For ImageNet, zero-shot and classifier accuracy show downstream task performance.

# E METRICS

## E.1 VLM ENCODER METRICS

**Recall@k.** Let $I, T \in R^{b \times d}$ be normalized embeddings of matching image and captions. Let $C = I \cdot T^{\top} \in [-1, 1]^{b \times b}$ be the matrix of cosine similarities between these embeddings. Let $l$ be the vector of ground truth matching labels : $l = [\![0, b-1]\!] \in \mathbb{N}^{b}$, and $\text{topk}^{(\text{img})} \in \mathbb{N}^{b \times k}$ be such that $\text{topk}_{i,j}^{(\text{img})}$ is the $j$-th best match for the image embedding $i$, and likewise for $\text{topk}^{(\text{txt})}$. We can now define $\text{recall@k} = \frac{1}{2b} \sum_i \left( \mathbf{1}_{i \in \text{topk}_i^{(\text{img})}} + \mathbf{1}_{i \in \text{topk}_i^{(\text{txt})}} \right)$. In other words, it measures the accuracy of the model at matching image-caption pairs. As this metric depends on the batch size $b$, we will fix it to 256 throughout this study.

**Zero-shot accuracy.** The goal of this metric is to measure the quality of multimodal embeddings through downstream task performance. In the context of image classification, text embeddings are

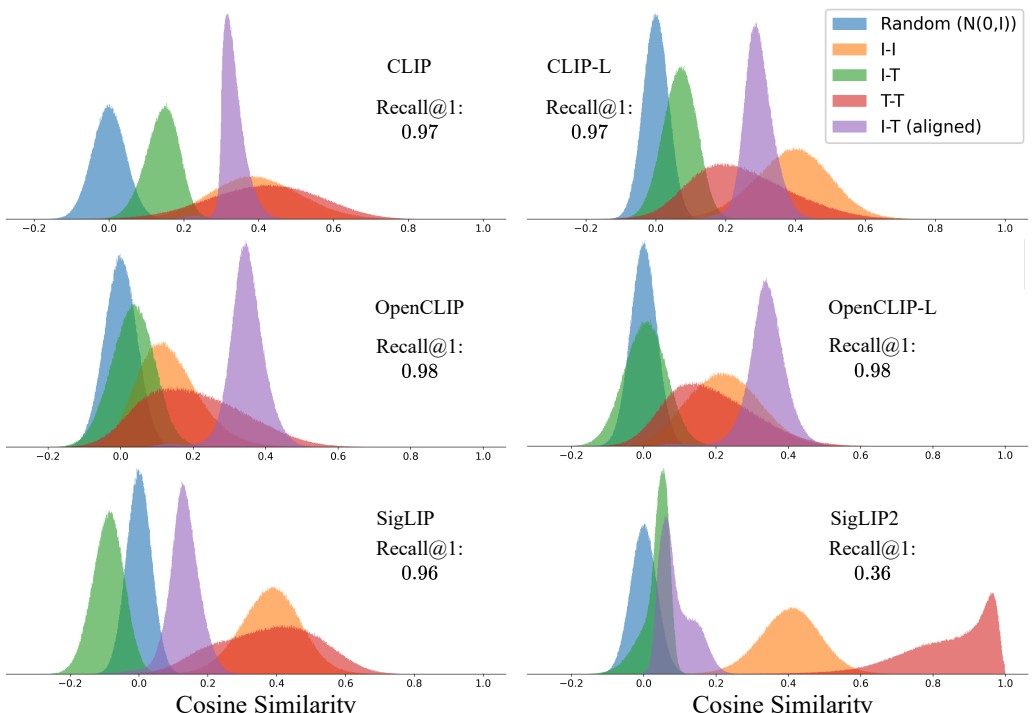

Figure 11: Histograms of cosine similarities between pairs of embeddings from the same modality (I : image, T : text), random pairs of opposing modality (I-T), and matching image-caption pairs (I-T (aligned)). Cosine similarities between samples from a white noise are indicated as a reference. We only consider LAION embeddings here. We indicate the corresponding recall@1 performance on each plot.

constrained to a small number of points, corresponding to simple descriptions of the class—e.g., "an image of a {class}". Following Radford et al. (2021), we generate about 80 text descriptions for each ImageNet class, embed them, take their average, and finally normalize this average to get the class classifier. By concatenating all of these embeddings, we get a linear classifier, and zero-shot accuracy is the accuracy of this classifier. As for recall, "@k" means that the prediction is correct if the correct class is among the $k$ best matches.

**Classifier Accuracy.** Alternatively, we train a linear classifier on the image embeddings instead of taking the one given by the embedding of these basic textual class descriptions. This metric doesn't consider the multimodal aspect of the embeddings anymore; it only considers whether the embeddings contain the information to classify.

**Retrieval.** We evaluate retrieval performance, another downstream task, on the FashionIQ dataset (Appendix J). To do so, we consider the average of recall@10 and recall@50, as in Wu et al. (2021). Instead of matching images $I$ and texts $T$, we now have queries $Q$ and targets $T$. Queries are functions of the source image's embedding and the relative caption's embedding, typically the normalized sum. Targets are the target image's embedding.

The goal is to quantify the quality of semantic vector arithmetic. Qualitatively, the embedding of an image of a red shoe $\boldsymbol{i}_{\mathrm{src}}$ plus the embedding of the relative description *"is not red but blue"* $\boldsymbol{\Delta}$ should match the embedding of an image of a blue shoe $\boldsymbol{i}_{\mathrm{src}} + \boldsymbol{\Delta} \sim \boldsymbol{i}_{\mathrm{trg}}$. Quantitatively, we evaluate both if our queries have high recall—meaning they can retrieve images of blue shoes—and whether they belong to the distribution of images—meaning they are actually embeddings of blue shoes—using the distance to the K-th Nearest Neighbor, a standard out-of-distribution (OOD) detection method (Sun et al., 2022; Yang et al., 2022).

### E.2 SAE METRICS

#### E.2.1 GENERAL METRICS

**Sparse Reconstruction.** These metrics measure reconstruction faithfulness and the sparsity of the codes. While a perfect reconstruction indicates the SAE likely fits noise, a faithful reconstruction of the activations is necessary to have captured the latent conceptual structure.

- $\text{MSE} = \mathbb{E}_i \left[ \|A_i - \widehat{A}_i\|_2^2 \right]$, where $\widehat{A}_i$ is the reconstruction of $A_i$ by the SAE. This is a direct measure of the reconstruction error, quantifying how closely the autoencoder reproduces the original activations.

- $R^2 = \mathbb{E}_i \left[ 1 - \frac{\|A_i - \widehat{A}_i\|_2^2}{\|A_i - \bar{A}\|_2^2} \right]$, where $\bar{A}$ is the average activation in $A$. The $R^2$ score is a measure of the explained variance. It is independent of the norm of the activations, focusing on the proportion of variance captured by the model on each instance.

- $L_0 = \mathbb{E}_i \left[ \|z_i\|_0 \right]$, where $z_i$ is the encoding of $A_i$ by the SAE, and $\| \cdot \|_0$ counts the number of nonzero elements. This metric captures the average sparsity level in the encoded representation.

- $L_1 = \mathbb{E}_i \left[ \|z_i\|_1 \right]$, where $z_i$ is the encoding of $A_i$. This is the average $L_1$ norm of the codes and serves as a differentiable proxy for $L_0$ sparsity. It penalizes large activations and encourages sparse representations, and is commonly used as a regularization term. In some architectures, a term similar to $L_1$ is included in the loss to promote sparsity, while in others, where sparsity is directly controlled, it is not necessary. We report it for all architectures for comparison.

**Consistency.** Measures whether the dictionary is well-grounded and functionally relevant.

- Interventions: $C-$insertion, $C-$deletion (Fel et al., 2023a) We include these metrics for completeness, even though they are not very useful in our case. For all data points, they sort active features by attribution and either insert or delete them one by one, measuring the downstream effect. Attribution is relative to a scalar, and we used the reconstruction error in this case. This is not very informative, as activation is then a very good proxy for attribution due to the near orthogonality of features. What these metrics tell us is essentially how fast the activation decreases if you sort features by activation. An arguably more useful way of computing these metrics would have been w.r.t. the cosine similarity between matching pairs or w.r.t. the recall. But again, it would not be very informative, and they would essentially become weaker versions of $\delta_{\text{r@1}}$.

- Stability measures the average distance between learned dictionaries under different random seeds. It is traditionally believed that lower is better, as it means the algorithm and its inductive biases solve the unidentifiability problem. However, it does not, by any means, imply that the found solution is "good" or is representative of the "true" conceptual structure.

**Structure in $D$.** Organization of the learned dictionary atoms.

- $\text{StableRank}(M) = \frac{\|M\|_F^2}{\|M\|_2^2}$. Here, $\|M\|_F^2 = \sum \sigma_i^2$ and $\|M\|_2^2 = \sigma_{\max}^2$, where $\sigma_i$ are the singular values of $M$. It gives both the number of significant singular values in $M$ and a robust rank estimation. Due to numerical and floating-point errors, almost all high-dimensional matrices are full rank.

- $\text{EffectiveRank}(M) = e^{H(p)}$, where $H(p) = -\sum p_i \log p_i$ is the entropy of the probability distribution $p$, and $p_i = \frac{\sigma_i}{\sum \sigma_j}$. It captures information-theoretic diversity in $M$, measuring the spread of the spectrum. It is the number of informative singular directions in the matrix. Both a $\text{StableRank}(M)$ and an $\text{EffectiveRank}(M)$ close to 1 suggest a few dominant directions, while values close to $\text{rank}(M)$ indicate all directions are equally important.

When analyzing $D$, higher stable and effective ranks are better, which suggests high diversity of concepts.

- Coherence $= \max_{i \neq j} D_i^\top D_j$ is the maximum cosine similarity between two dictionary atoms. A score close to 1 indicates possibly redundant concepts, while lower scores tend to reflect a better spread of concepts. In practice, this metric seems to be a measure of the inductive bias against learning close, undistinguishable features, with all SAEs but MP in Appendix A having this bias and achieving similar scores, and MP not having this bias and having a much higher score. See Appendix F for more details on MP.

**Structure in $Z^\top Z$.** In this case, we consider $Z \in \mathbb{R}^{N \times K}$ with normalized columns, to focus on co-activation structure, not just activation magnitude.

- Connectivity $= 1 - \frac{1}{K^2} \left\| Z^\top Z \right\|_0$ counts the number of unique co-activations among all possible pairs. Higher connectivity scores suggest a wide range of possible meaningful concept combinations, while low connectivity implies a sparse graph of concept combinations, with possibly more structure in this second case.
- StableRank and EffectiveRank measure the structure in $Z^\top Z$, with low scores indicating highly structured co-activations, with the presence of groups or blocks of concepts behaving in similar manners. High scores suggest little to no structure.
- NegativeInterference $= \left\| \mathrm{ReLU} \left[ -(Z^\top Z) \odot (DD^\top) \right] \right\|_2$ quantifies how much concepts tend to cancel each other out.

### E.2.2 MODALITY SPECIFIC METRICS

In addition to these general metrics, we introduce multimodality-specific metrics, motivated by the Iso-Energy Assumption and the phenomenology of VLM embeddings. We report:

1. *Probing accuracy* $p_{\mathrm{acc}}$, which quantifies how the geometric organization of dictionary atoms aligns with known structures in latent space.

2. *Functional alignment* $\rho$, an instance-level ratio indicating whether cross-modal alignment is carried by *bimodal* rather than *unimodal* features.

3. *Functional and Distributional Agreement* (FDA), a distribution-level analogue of $\rho$ that checks consistency of this functional role across populations.

4. $\delta_{\mathrm{r@1}}$, an interventional measure of how much retrieval recall degrades when *unimodal* atoms are ablated.

Together, these four views (geometric, functional, distributional, and causal) consistently probe complementary aspects of how features support cross-modal alignment. While $p_{\mathrm{acc}}$ gives a purely geometric measure of consistency between structures in concepts and embeddings, $\rho$ and FDA link the geometric aspect of alignment to the functional behavior of each feature. $\rho$ and FDA are two sides of the same coin: while $\rho$ operates at the instance level, capturing how cross-modal alignment is distributed across features functionally, FDA operates at the distribution level and shows whether this functional role is consistent with the global geometry of the two modalities, in a way that is more consistent with the Iso-Energy Assumption. $\delta_{\mathrm{r@1}}$ complements these functional views by providing a causal perspective on whether *bimodal* features alone are sufficient to maintain alignment.

**Energy.** Energy $E$ gives a measure of the importance of each feature in the dictionary. See Figure 16a for a detailed distribution of energy across features and modality.

$$\forall \mathrm{d} \in \mathfrak{D} \quad E^{(\mathrm{d})} = \mathbb{E}\left( \boldsymbol{\psi}^{(\mathrm{d})}(\boldsymbol{X})^2 \mid \boldsymbol{X} \in \mathcal{X}^{(\mathrm{d})} \right) \qquad E = \frac{1}{|\mathfrak{D}|} \sum_{\mathrm{d} \in \mathfrak{D}} E^{(\mathrm{d})} \in \mathbb{R}^K \qquad (5)$$

where $\psi$ is the composition of the VLM and the SAE. Let $\mathbf{a} := \frac{E^{(\mathrm{img})}}{\left\| E^{(\mathrm{img})} \right\|_1}$ and $\mathbf{b} := \frac{E^{(\mathrm{txt})}}{\left\| E^{(\mathrm{txt})} \right\|_1}$ be probability distributions for the image and text domains, respectively.

**Modality Score.** Before turning to the metrics themselves, we introduce the modality score, and a binary mask indicating whether a feature is *bimodal*. The *modality score* of feature $i$ is a direct measure of the component-wise Iso-Energy Assumption: $\mu_i = \frac{E_i^{(\mathrm{img})}}{E_i^{(\mathrm{img})} + E_i^{(\mathrm{txt})}} \in [0, 1]$. Values $\mu_i \approx 1$ or $\mu_i \approx 0$ correspond to image-only and text-only features, while $\mu_i \approx 0.5$ indicates balanced

activation energy across modalities. We refer to the former as *unimodal* features and the latter as *bimodal* features. To make this distinction operational, we introduce a binary mask $\boldsymbol{\delta}_i$ indicating whether feature $i$ is classified as *bimodal*. In practice, $\boldsymbol{\delta}_i$ is obtained by thresholding $\mu_i$ within an interval $[\tau, 1-\tau]$ around 0.5, with $\tau = 0.05$. We use a bridge-based consistency criterion to define $\tau$, which we introduce later.

**Linear Structure.** We evaluate how the geometric organization of dictionary atoms aligns with the known structure of the modality gap. In Figure 2, a linear projection of feature vectors reveals three distinct clusters corresponding to image-only, text-only, and *bimodal* features. This geometry of concept organization closely aligns with the geometry of embeddings and the modality gap : image–only features are aligned with the image cone, similarly for text, while *bimodal* features are orthogonal to modality information. We define *probing accuracy* $p_{\mathrm{acc}}$ to quantify how well the geometric structure of concepts aligns with embedding structure and the modality gap. High $p_{\mathrm{acc}}$ indicates strong alignment.

Each feature $D_i$ is considered as a linear probe classifying embeddings. *Unimodal* features are expected to be highly predictive, yielding high accuracy, whereas *bimodal* features should be domain-agnostic, yielding an accuracy near 0.5. The score $s_i$ of *unimodal* feature $i$ is simply its accuracy *acc*. For *bimodal* features, $s_i = 1 - 2 * (\max(\mathrm{acc}, 1 - \mathrm{acc}) - 0.5)$ is close to 1 if they are consistently anti-informative (i.e. good at being bad), and close to 0 otherwise. Finally, $p_{\mathrm{acc}}$ is computed as the energy-weighted average of the $s_i$ over all features: $p_{\mathrm{acc}} = \frac{\sum_i E_i \cdot s_i}{\sum_i E_i}$.

**Bridge.** Some features naturally act as *bridges* between modalities, carrying information that contributes to cross-modal alignment. Intuitively, these are the *bimodal* features: they are simultaneously active in both image and text embeddings and facilitate matching across modalities. Following Papadimitriou et al. (2025), we define the *bridge* matrix $B_W$ as $B_W = W \odot (D \cdot D^\top) \in \mathbb{R}^{K \times K}$ where $W \in \mathbb{R}^{K \times d}$ is a weight matrix. They used $B_\Sigma$, where $\Sigma$ is the cross-covariance of activations across image-text pairs from a joint empirical distribution $\gamma$ (Equation (6)).

$$\Sigma = \mathop{\mathbb{E}}_{(\boldsymbol{X}, \boldsymbol{X}') \sim \gamma} \left( \boldsymbol{\psi}^{(\mathrm{img})}(\boldsymbol{X})^\top \boldsymbol{\psi}^{(\mathrm{txt})}(\boldsymbol{X}') \right) \tag{6}$$

$$\Gamma = \arg\min_\gamma \| \gamma \odot C \|_F \qquad \text{s.t.} \quad \gamma \mathbf{1} = \mathbf{a}, \tag{7}$$

$$\mathbf{1}^\top \gamma = \mathbf{b}^\top, \ \gamma \geq 0 \tag{8}$$

We also consider $B_\Gamma$, where $\Gamma$ is the optimal transport plan (Equation (7)) between the image- and text-weighted dictionaries, using weights $\mathbf{a}$ and $\mathbf{b}$, and a cost matrix $C = 1 - D \cdot D^\top$ (equivalently, both cosine distance and half squared Euclidean distance since atoms are unit norm). While $B_\Sigma$ captures instance-level co-activation patterns under the contrastive objective, $B_\Gamma$ captures alignment at the distribution level, making it more consistent with the Iso-Energy assumption and robust to instance-level mismatches. We show how to chose $\tau$ based on $B_\Sigma$ in Appendix E.2.2.

**Functional Alignment.** $B_\Sigma$ captures *functional alignment*, the combined functional similarity and geometric alignment of features across modalities. It reflects each feature pair's contribution to cross-modal alignment under the contrastive objective, with its entries providing pairwise alignment score. Thus we define $\rho$ the ratio between total $B_\Sigma$ bridge score adjacent to *bimodal* features —i.e. for all pairs $(i, j)$ where at least one of $i$ or $j$ is a *bimodal* feature— and that adjacent only to *unimodal* features —i.e. both $i$ and $j$ are *unimodal*. $\rho > 1$ means functional alignment is primarily done by *bimodal* features; $\rho < 1$ means *unimodal* pairs spuriously carry it.

**Distributional Alignment.** In contrast, $B_\Gamma$ captures *distributional alignment* between modalities. Its total mass is $1 - c$, where $c$ is the transport cost. The entries of $B_\Gamma$ reveal which features contribute to balancing or imbalancing the alignment. A high $B_{\Gamma,i,j}$ indicates features $i$ and $j$ jointly contribute to aligning the two distributions. A low score is harder to interpret: it may reflect features contributing to imbalance (if $\Gamma_{i,j} \neq 0$) or unmatched pairs by the transport plan. *Bimodal* features should belong to both modalities' distribution (particularly matching mostly to themselves) and thus drive distributional alignment. On the other hand, *unimodal* features should be specific to it's corresponding

modality's distribution and thus should not participate in the overall distributional alignment. This motivates the **Functional and Distributional Agreement (FDA)** ratio:

$$\text{FDA} = \frac{\alpha}{\varepsilon} \frac{1 - \varepsilon}{(1 - c) - \alpha}, \text{ where } \alpha = \sum_{i,j|\boldsymbol{\delta}_i \vee \boldsymbol{\delta}_j} B_{\Gamma,i,j} \text{ and } \varepsilon = \frac{\sum_{i|\boldsymbol{\delta}_i} E_i}{\sum_i E_i} \tag{9}$$

Here, $\alpha$ is the sum of bridge scores adjacent to a *bimodal* feature, and $\varepsilon$ is the proportion of energy in *bimodal* features. FDA is a mirror of the instance–level $\rho$. It summarizes the impact of *bimodal* and *unimodal* features on the distributional alignment between the two modalities. The higher the FDA, the better the dictionary is at capturing features whose functional behavior consistently matches their geometric distribution across modalities. Indeed, *bimodal* features should always carry most of the alignment *relative to their overall energy*. A low FDA means either *bimodal* features fail to align distributions properly or *unimodal* features are spuriously filling this role, indicating a breakdown in the shared conceptual structure.

**Intervention.** After these two functional measures, we turn to the causal behavior of *bimodal* feature, asking if they alone are sufficient to explain the recall performance of the embeddings. To do so, we define $\delta_{\text{r@1}}$ as the difference in recall between embeddings reconstructed with the full dictionary and those reconstructed by filtering out *unimodal* features. Let $A \in \mathbb{R}^{N \times d}$ be a set of activations, $Z \in \mathbb{R}^{N \times K}$ the set of sparse codes obtained by encoding $A$ with our learned encoder, $\boldsymbol{\delta} \in \{0,1\}^{N \times K}$ be a binary mask filtering out *unimodal* features, broadcasted across the rows of $Z$. Then, let $\widehat{A} := ZD$ be the set of reconstructed activations, and $\widetilde{A} := (Z \odot \boldsymbol{\delta})D$ be the set of *bimodal* activations. We finally measure $\delta_{\text{r@1}}$ as the difference in recall between the embeddings given by $\widehat{A}$ and $\widetilde{A}$.

**Thresholding $\mu$.** These four metrics depend on the threshold chosen to decide which features are *bimodal*, text-only, and image-only. See Figure 12 for a visualization of $\rho$ across the full range of possible thresholds. This figure also illustrates how to choose the threshold, depending on the region's size, involving bridges between two image-only or two text-only features. As soon as this region becomes significant, *unimodal* features can't be considered unimodal anymore. As we can see, our results are not very sensitive to the threshold. This is explained by the fact that most modality scores are extreme, either close to 0, 0.5 or 1. This is best visualised through the domain-wise distribution of energy (Figure 16a), where we can see clearly defined modes that are not sensitive to the threshold.

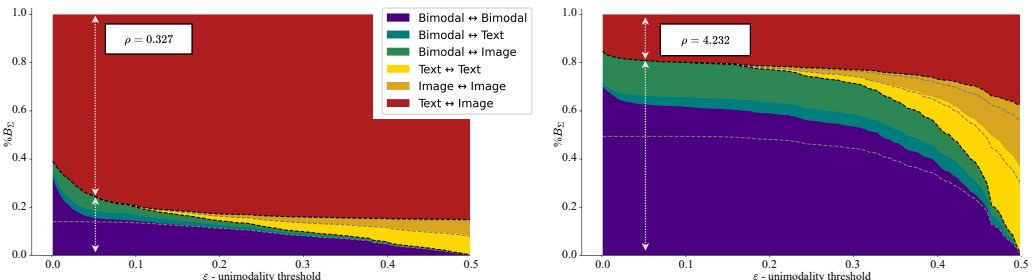

Figure 12: Bridge $B_\Sigma$ mass repartition between *bimodal*, image-only, and text-only features, depending on the unimodality threshold.

Figure 13 shows an illustration of the FDA score, with both an example of a high score on a SAE-A where we can clearly see that *bimodal* features are very stable while image-only are transported to text-only features at considerable cost, and a low score on a classical SAE, where there is no clear dynamic between all three types of features, especially in the center cluster where all three types seem to be mixed.

We can see in Figure 14 the details of the $p_{\text{acc}}$ score distribution across each feature, both for an SAE and a SAE-A.

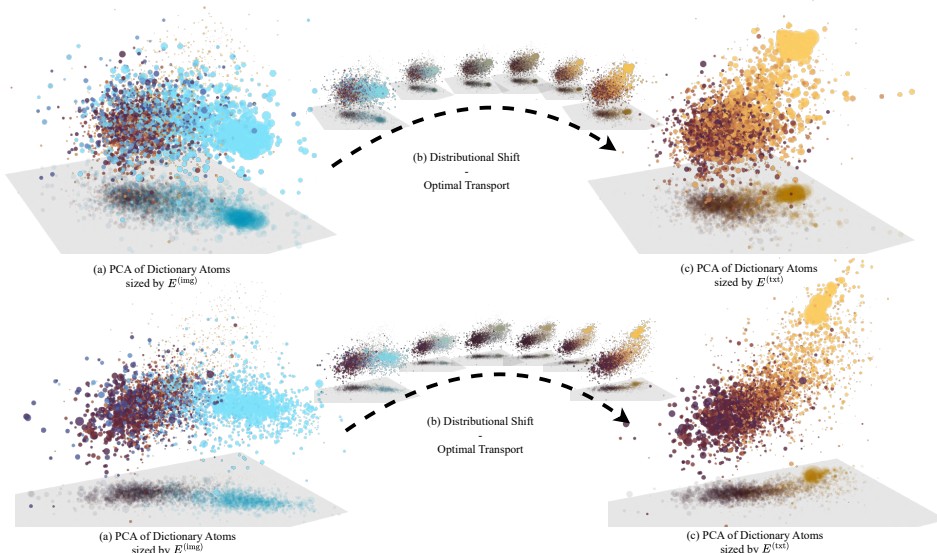

Figure 13: **Features of SAE-A (Bottom, FDA = 4.559) functionally behave in accordance with their geometric repartition, which is less the case in SAE (Top, FDA = 2.630)**. The left and right plots show 3D projections of the feature distributions, colored by modality score, sized by energy on the image (*left*) and text (*right*) domains. The center part displays optimal transport between the two domains' distribution, illustrating the FDA score.

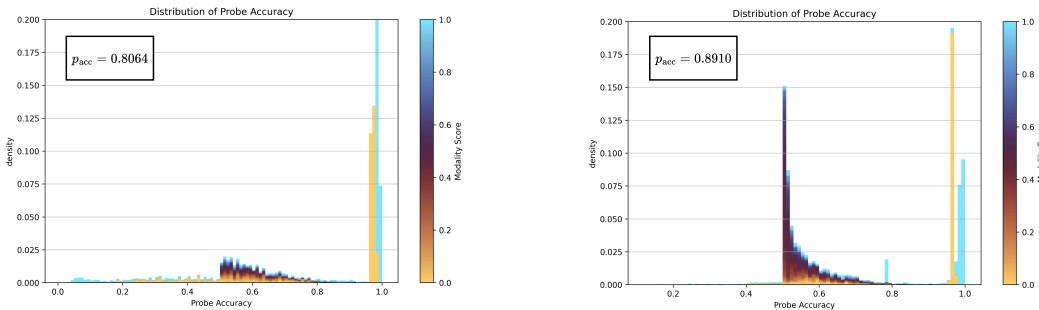

Figure 14: **Features in SAE-A (right) outperform those in SAE (left) at geometrically aligning with modality-specific or agnostic information**. Distribution of probing accuracy for each feature based on its modality score. Each feature is evaluated as a probe classifying the modality of embeddings. *Unimodal* features should have an accuracy of 1, while *bimodal* features should have an accuracy of 0.5. $p_{\text{acc}}$ summarizes that in a single metric—higher is better.

### E.3 OPTIMIZING FOR $p_{\text{acc}}$

Let's dive into why optimizing for $\mathcal{L}_{\text{align}}$ does not optimize for $p_{\text{acc}}$. Before presenting the technical reasoning, it is important to clarify the motivation for making this distinction.

We consider each feature vector $D_i$ as a probe on embeddings $A$ by computing the dot product between $D_i$ and every embedding $A_j$ : scores $= A \cdot D_i \in \mathbb{R}^N$, and then taking binary classification predictions as preds $= \mathbf{1}_{scores>0} \in \{0,1\}^N$. Given some desired labels $l \in \{0,1\}^N$, we can thus compute the accuracy of $D_i$. Precisely, the scores are corrected to have a mean of $0$, which is equivalent to, and done by, centering $A$. The reason for this is that we know exactly half of the labels are $0$s and half are $1$s.

Let $l^{(\text{img})}$ be labels indicating which embeddings are from the image domain. For an image-only feature, with $\mu > 1-\tau$, we take $l = l^{(\text{img})}$. For a text-only feature, with $\mu < \tau$, we take $l = 1-l^{(\text{img})}$. Finally, as explained above, for *bimodal* features, we take $l = l^{(\text{img})}$ and apply a transformation to

the probe's accuracy such that the resulting score is close to 0 when the feature acts as a reliable (text or image) classifier, and approaches 1 when the classification accuracy is near chance (i.e., 0.5): $\mathrm{acc}_{\mathrm{bimodal}} = 1 - 2 * (\max(\mathrm{acc}, 1 - \mathrm{acc}) - 0.5)$.

If we consider the distributions of $I \cdot D_i$ and $T \cdot D_i$, it can appear that $\mathcal{L}_{\mathrm{align}}$ pushes these distributions to be equal, by enforcing an equal activation on both elements of a matching image-caption pair. However, this is only true when $z_i > 0$ for at least one of the image or the caption. In other words, if you consider only a subset of $I$ and $T$ where $D_i$ is active, then it is true that $\mathcal{L}_{\mathrm{align}}$ penalizes distribution mismatches.

Nevertheless, as features are sparse and have very low frequency, this effect is negligible on the total distributions, as illustrated by Figure 15.

Moreover, all of that is applicable only for *bimodal* features. *Unimodal* features are not affected at all by this effect, and yet, we observe that in SAE a significant amount of *unimodal* features act as classifiers for the opposite modality, which almost completely disappears in SAE-A. This suggests that when training with $\mathcal{L}_{\mathrm{align}}$, features become more organized overall.

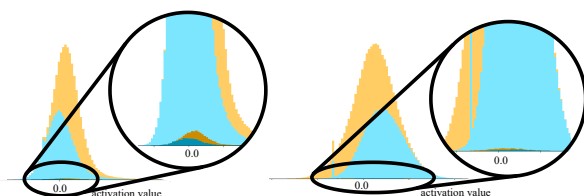

Figure 15: Distribution of $\mathrm{scores} = A \cdot D_i$ for a high energy *bimodal* feature (**left**) and a low energy *bimodal* feature (**right**). Blue: image scores $I \cdot D_i$. Yellow: text scores $T \cdot D_i$. Dark: subset of image-caption pairs where $D_i$ is active in at least one of them. Light: all other pairs. The implicit optimization for $p_{\mathrm{acc}}$ happens only in the dark regions.

## F  MP-SAE HIGH ENERGY "BIASES"

In this section, we investigate high-energy *unimodal* features, circled in Figure 16a. We find that they are all extremely close to one another, with an average cosine similarity above 0.9 (Figure 16b), never co-activate, and have a total frequency of 0.5. As they are *unimodal*, this means that they are, in fact, small variations around a *unimodal* bias. These results hold for all the models we tested, and for all the MP-SAEs we trained.

We find that, while some of these features do not seem to be trivially interpretable by looking at their most activating examples, some are surprisingly interpretable (Figures 19 and 21).

Figure 17 additionally shows the distribution of energy and frequency across features.

## G  FEATURE GALLERY

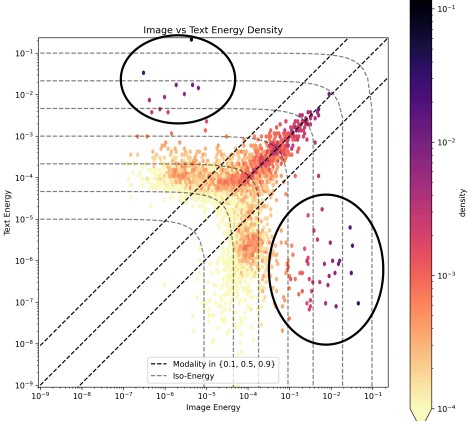

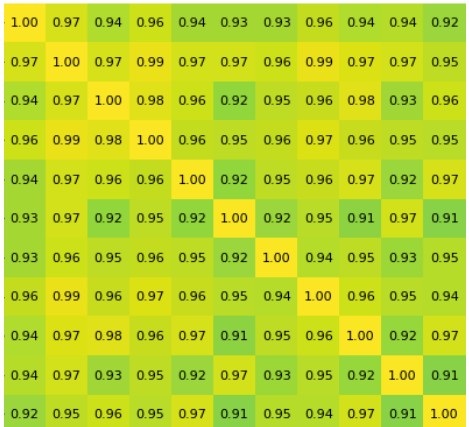

(a) 2D histogram of feature energy across both the image and text domains. We circled high-energy *unimodal* features for the analysis of Appendix F.

(b) Matrix of pairwise cosine similarity between high-energy text-only features. Average cosine similarity: 0.9555 Total frequency: 0.461

Figure 16: **Left:** energy distribution across domain, **right:** pairwise cosine similarity between circled features.

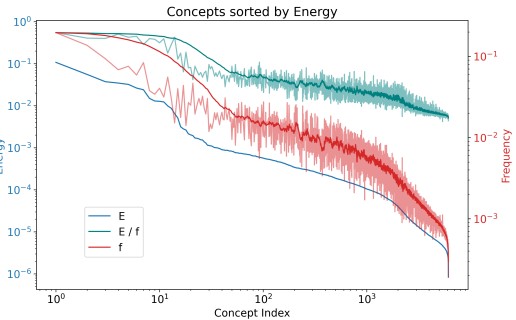

Figure 17: Energy and frequency distribution across features, sorted by energy. The $E/f$ curve gives an indication of the activation magnitude when the feature is active. Apart from the first few features, it appears that this quantity is roughly constant across all features.

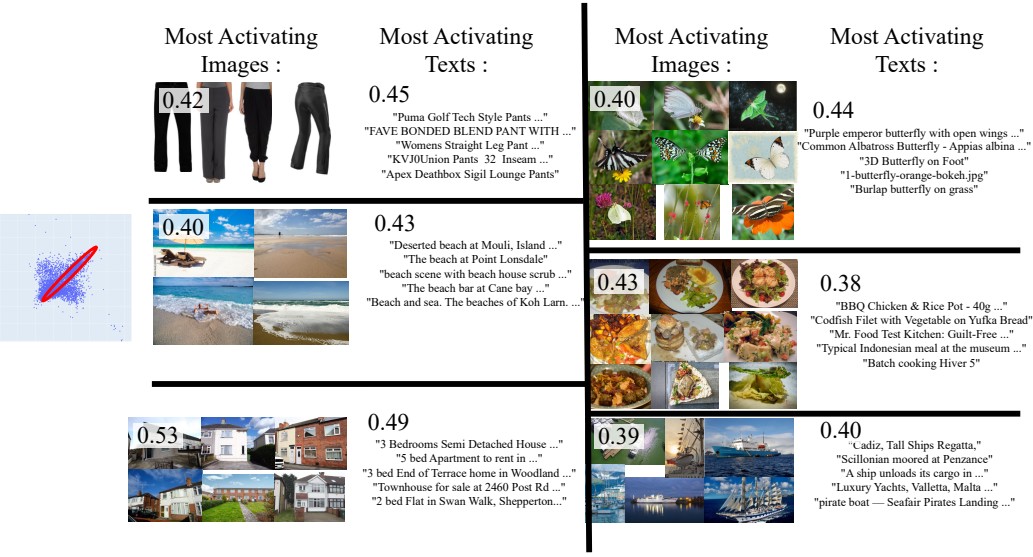

Figure 18: Examples of high and low-energy *bimodal* features' most activating samples. For each feature, we indicate its average activation on the selected samples.

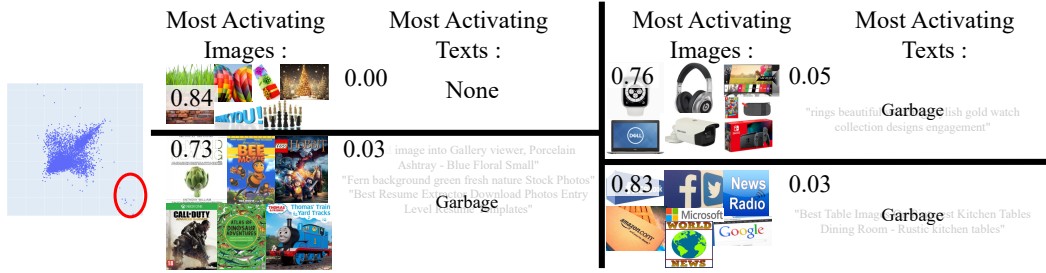

Figure 19: Examples of image-only features' most activating samples. For each feature, we indicate its average activation on the selected samples. We can see an a priori uninterpretable feature, then a feature for electronic devices, covers, and logos. These features are almost never active in texts, and when they are, they have very small activation and no apparent pattern.

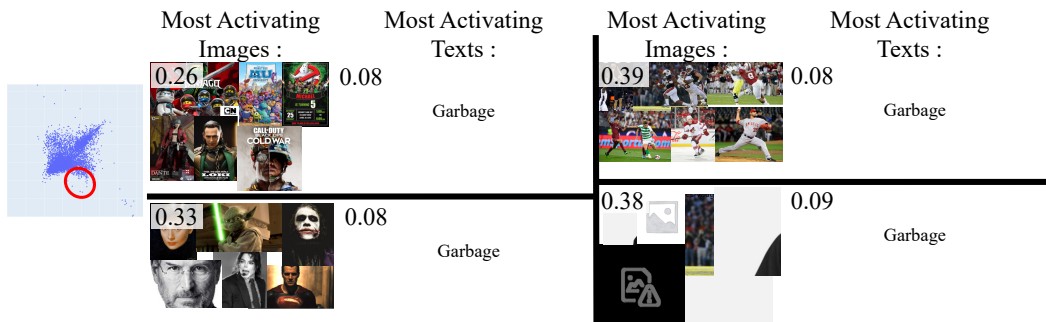

Figure 20: Examples of image-only features' most activating samples. For each feature, we indicate its average activation on the selected samples. We can see a feature for covers, sports, famous figures, and badly cropped images. These features are almost never active on texts, and when they are, it's with very small activation and without any apparent pattern.

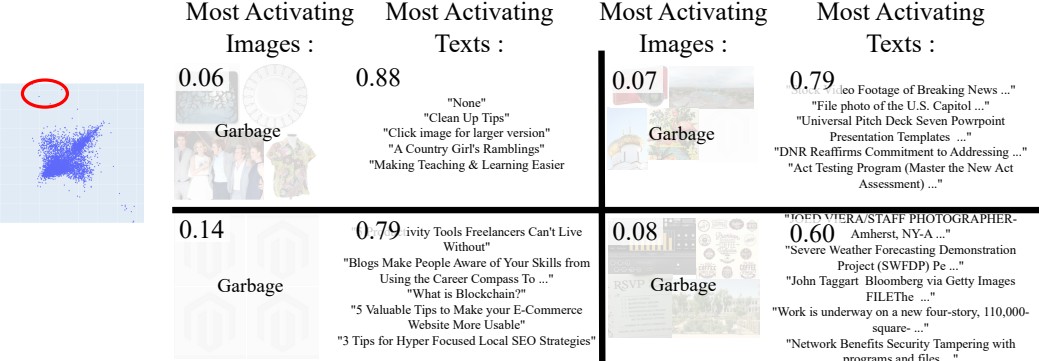

Figure 21: Examples of text-only features' most activating samples. For each feature, we indicate its average activation on the selected samples. We can see an a priori non-interpretable feature, then a feature for news, scam influencers, and news again. These features are almost never active on images, and when they are, it's with very small activation and without any apparent pattern.

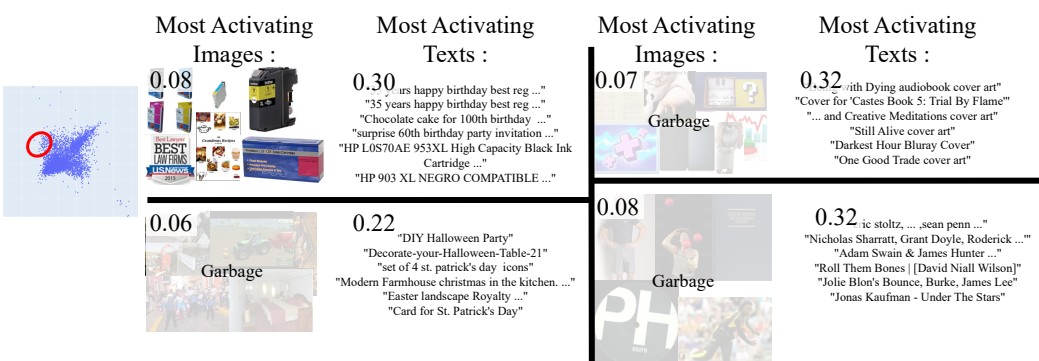

Figure 22: Examples of text-only features' most activating samples. For each feature, we indicate its average activation on the selected samples. We can see a feature for ink cartridges and birthdays, cover art, traditional holidays, and name-dropping. These features are almost never active on images, and when they are, they have very small activation and no apparent pattern, except for the cartridges and birthday one, which also activate with low energy on similar images.

# H SEMANTIC VECTOR ARITHMETIC.

After training SAEs with desirable properties, based on our hypotheses on the representations of VLM dual encoders, we now ask whether they can be used to improve their performance on downstream tasks. To do so, we select both zero-shot accuracy, measured on ImageNet in Appendix I as part of the modality gap experiment, and retrieval, measured in this section on FashionIQ. See Appendix J for details on this dataset.

**Notations.** Let $Q \in R^{N \times d}$ be query vectors and $T \in \mathbb{R}^{N \times d}$ be target vectors. The task consists in maximizing average recall on batches $b$ of cosine similarity matrix $Q_b \cdot T_b^\top$. $T$ are embeddings of target images, and $Q$ will involve $I_{\text{src}}$, source image embeddings, and $\Delta$, embeddings of relative captions. Let $Z_T$, $Z_{I_{\text{src}}}$ and $Z_\Delta$ be the sparse codes of $T, I_{\text{src}}$ and $\Delta$ given by our SAE.

Let $\widehat{T} := Z_T D$ be the set of reconstructed activations, and $\widetilde{T} := (Z_T \odot \boldsymbol{\delta})D$ be the set of *bimodal* activations (recall that $D \in \mathbb{R}^{K \times d}$ is the dictionary and $\boldsymbol{\delta} \in \{0, 1\}^K$ is the binary mask selecting *bimodal* features, and broadcasted to the size of the codes). Similarly, let $\widehat{I}_{\text{src}}$, $\widetilde{I}_{\text{src}}$, $\widehat{\Delta}$ and $\widetilde{\Delta}$ be reconstructions and *bimodal* versions of $I_{\text{src}}$ and $\Delta$.

**Queries.** We define the following queries. First, we use $Q_{\widehat{I}_{\text{src}}}^{\text{baseline}} := \widehat{I}_{\text{src}}$ and $Q_{\widehat{\Delta}}^{\text{baseline}} := \widehat{\Delta}$ as lower bounds for any other queries. Then, let $Q := \widehat{I}_{\text{src}} + \widehat{\Delta}$ be the classical way of doing semantic vector arithmetic, and $Q_{\text{SAE}} := \widehat{I}_{\text{src}} + \widetilde{\Delta}$ be arithmetic restricted on *bimodal* features.

See Figure 5 for a visual representation of all vectors involved. Note that queries are scaled to end up on the unit sphere on this representation, the actual result of the addition is indicated with a cross.

**Retrieval.** We find that $Q_{\text{SAE}}$ systematically outperforms $Q$ in terms of retrieval recall, across the six models we tested (Table 8).

**Out of Distribution.** We also measure the out of distribution (OOD) score of each of our four queries distribution to the target distribution. The idea is to measure whether our queries actually match targets or are just vectors that happen to be able to do retrieval.

We use the distance between $Q_i$ and it's K-th nearest neighbor in $\widehat{T}$— similarly for $T_i$—then we find the optimal threshold on the distance to classify $Q$ vs $\widehat{T}$. **The final OOD score of Q is the accuracy of this classifier**. This approach is popular in the OOD literature (Sun et al., 2022; Yang et al., 2022). We typically use $K = 10$. We repeat that procedure for $Q_{\text{SAE}}$ and the two baselines.

We expect $Q_{\widehat{I}_{\text{src}}}^{\text{baseline}}$ to have the exact same distribution as $\widehat{T}$ and $Q_{\widehat{\Delta}}^{\text{baseline}}$ to be extremely out of distribution, due to the modality gap. Also due to the modality gap, we expect $Q$ to be completely OOD. Intuitively, if you only consider *unimodal* biases, $Q$ will have the mean of both. As biases are the strongest components of all embeddings, this means that $Q$ will end up in a cone disjoint from both the image and text embedding distribution. In practice, there are a lot of other *unimodal* features that are not biases. Thus, by intervening only on shared representations, we should have $Q_{\text{SAE}}$ end up in the same distribution as $\widehat{T}$.

Results are available in Figure 23 and Table 7. We can see that, as expected, restricting the intervention to shared representations reduces drastically the difference in query and target distribution.

| OOD score ($\downarrow$) | $Q$ | $Q_{\text{SAE}}$ |
|---|---|---|
| **CLIP** | 0.97 | **0.63** |
| **CLIP-L** | 0.95 | **0.76** |
| **OpenCLIP** | 0.86 | **0.68** |
| **OpenCLIP-L** | 0.87 | **0.72** |
| **SigLIP** | 0.99 | **0.70** |
| **SigLIP2** | 0.99 | **0.61** |

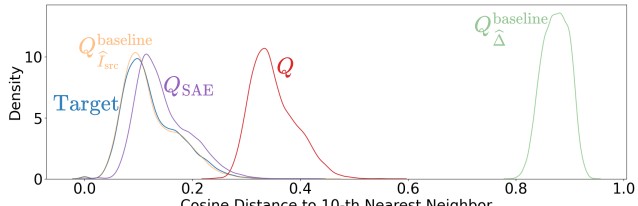

Table 7: OOD score between the query distribution and target distribution. Across all six models, our concept based queries $Q_{\text{SAE}}$ outperforms the classical queries $Q$.

Figure 23: Histograms of distances between queries and their 10th nearest neighbor in the target distribution. Our $Q_{\text{SAE}}$ are close to the target distribution while classical $Q$ are not. $Q_{\widehat{\Delta}}^{\text{baseline}}$ and $Q_{\widehat{I}_{\text{src}}}^{\text{baseline}}$ are, as expected, perfectly OOD due to the modality gap, and perfectly in distribution as source and target image distribution are the same.

| $\frac{1}{2}(\text{r@10} + \text{r@50})$ ($\uparrow$) | **CLIP** | **CLIP-L** | **OpenCLIP** | **OpenCLIP-L** | **SigLIP** | **SigLIP2** |
|---|---|---|---|---|---|---|
| $Q_{\widehat{\Delta}}^{\text{baseline}}$ | 0.499 | 0.536 | 0.610 | 0.630 | 0.601 | 0.611 |
| $Q_{\widehat{I}_{\text{src}}}^{\text{baseline}}$ | 0.567 | 0.630 | 0.647 | 0.669 | 0.686 | 0.683 |
| $Q$ | 0.681 | 0.743 | 0.788 | 0.818 | 0.794 | 0.787 |
| $Q_{\text{SAE}}$ | **0.683** | **0.750** | **0.790** | **0.820** | **0.812** | **0.798** |

Table 8: Retrieval recall on FashionIQ for our two baselines and two queries. Across all six models, our concept based queries $Q_{\text{SAE}}$ outperforms the classical queries $Q$.

## I    REMOVING THE MODALITY GAP

We investigate two methods and two baselines for removing the modality gap between $I$ and $T$. We measure the modality gap using the difference in mean (DiM) and the Wasserstein distance $\mathcal{W}$, and consider the performance in recall on LAION. We also include COCO and ImageNet to study the transferability of our methods on unseen data distributions, and on unseen tasks.

**Notations.**    Let's first recall some notations. Let $I$ and $T$ denote image and text embeddings, with respective mean $\mu_I$ and $\mu_T$. Let $Z_I$, $Z_T$ be the codes of $I$, $T$ given by our SAE. Let $D$ be the dictionary, and $\boldsymbol{\delta} \in \{0, 1\}^K$ be the binary mask selecting *bimodal* features, and broadcasted to the size of the codes. Let $\widehat{I} := Z_I D$ be the set of reconstructed activations, and $\widetilde{I} := (Z_I \odot \boldsymbol{\delta})D$, and similarly for $\widehat{T}$ and $\widetilde{T}$.

Our first method comes from the analysis of our SAEs : we remove *unimodal* concepts, ending up with $\widetilde{I}$ and $\widetilde{T}$. This is the only method that doesn't enforce the mean of the two distribution to exactly match. Instead, we rely on our SAEs to capture all shared representations. As a baseline we propose to consider the centered versions of the data $I_{\text{center}} := \widehat{I} - \boldsymbol{\mu}_{\widehat{I}}$ and $T_{\text{center}} := \widehat{T} - \boldsymbol{\mu}_{\widehat{T}}$. The motivation for this baseline is the observation in Appendix F that *unimodal* features are predominantly small variations around a unique direction, except for small energy ones which are orders of magnitude less energetic.

For the above baseline and the next two methods, we use reconstructed embeddings instead of the original ones for fairness of comparison, as SAEs never perfectly reconstruct activations. Additionally, for the means, we use the means on the LAION dataset even on COCO and on ImageNet, also for fairness, as SAEs were trained on the LAION dataset.

The second method is called *embedding shift* in Liang et al. (2022), and it essentially consist in the following transformation : $I_{\text{shift}} := \widehat{I} - \boldsymbol{\mu}_{\widehat{I}} + \frac{\boldsymbol{\mu}_{\widehat{I}} + \boldsymbol{\mu}_{\widehat{T}}}{2}$ and $T_{\text{shift}} := \widehat{T} - \boldsymbol{\mu}_{\widehat{T}} + \frac{\boldsymbol{\mu}_{\widehat{I}} + \boldsymbol{\mu}_{\widehat{T}}}{2}$. We propose the following baseline for this method : we replace the average of the means by a random direction $\boldsymbol{r}$, sampled uniformly from the unit sphere—$I_{\text{rand}} := \widehat{I} - \boldsymbol{\mu}_{\widehat{I}} + \boldsymbol{r}$ and $T_{\text{rand}} := \widehat{T} - \boldsymbol{\mu}_{\widehat{T}} + \boldsymbol{r}$.

These two methods are not comparable to the first two in terms of modality gap, as even if the DiM is similar, the Wasserstein distance $\mathcal{W}$ behaves very differently. Indeed, in the first two methods, data is centered around $0$ and is roughly uniformly spread on the unit sphere. On the other hand, by manually adding a mean, these methods concentrate the data on a narrow cone, significantly reducing the $\mathcal{W}$. We illustrate these four methods on synthetic 3D points in Figure 24.

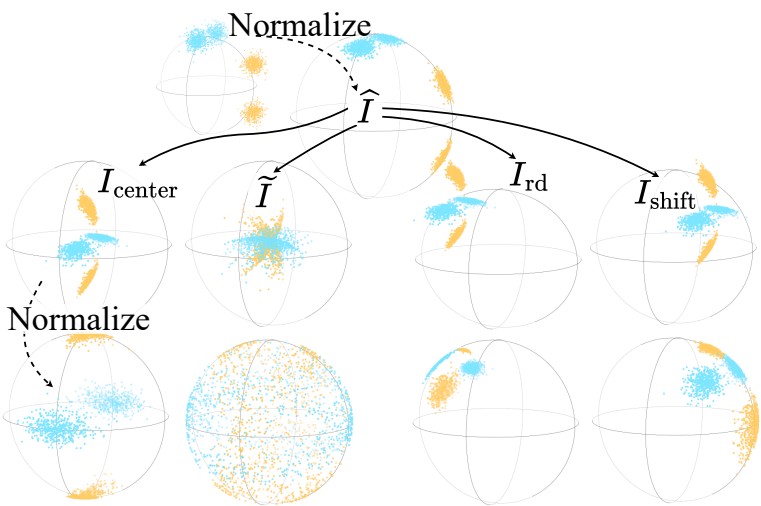

Figure 24: Illustration of the four methods to remove the modality gap on synthetic data points.

Results are compiled in Tables 9 to 11. We can see that the shift method introduced in previous work is by far the worst, significantly reducing recall performances, then comes the random baseline. The mean baseline outperforms the concept-based approach in terms of recall, but it does not close the modality gap.

We also indicate in Figure 25 the KNN distance histograms for our four methods, following the OOD detection method described in Appendix H, for the CLIP-L model on the LAION dataset. We take the image distribution as the reference one, and the text distribution as the "OOD" one. This illustrates a bit more precisely the effect of all methods on the distribution of embeddings. We can see that our method is the only one that closes the gap by merging the two distributions.

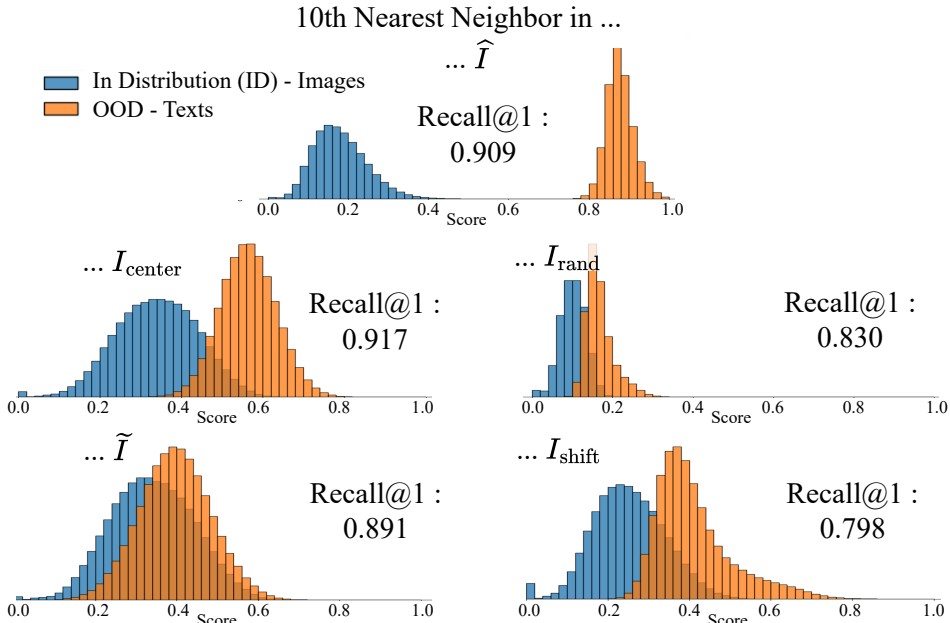

Figure 25: Histogram of distances from each image (ID) and caption (OOD) embedding to it's 10th nearest neighbor in the corresponding distribution of image embeddings. We use CLIP-L and the LAION dataset. As explained in Appendix I, the shift method and our concept based method can't be compared in terms of distances, they can only be compared to their respective baselines. Indeed, distances in the random baseline and shift method are artificially small, but we can see from these histograms that the two distributions can still be very well separated. They are however comparable in histogram separation and recall, which we indicate. **Our concept based method is the only one to truly close the gap by matching the image and text distribution.**

|  | CLIP | CLIP-L | OpenCLIP | OpenCLIP-L | SigLIP | SigLIP2 | $\Delta$ from $(\widehat{I}, \widehat{T})$ |
|---|---|---|---|---|---|---|---|
| DiM |  |  |  |  |  |  |  |
| $(\widehat{I}, \widehat{T})$ | 0.723 | 0.707 | 0.520 | 0.633 | 0.969 | 1.079 | / |
| $(I_{\text{center}}, T_{\text{center}})$ | 0.009 | 0.010 | 0.012 | 0.011 | 0.009 | 0.007 | -0.762 |
| $(\widetilde{I}, \widetilde{T})$ | 0.040 | 0.054 | 0.062 | 0.071 | 0.028 | 0.054 | -0.720 |
| $(I_{\text{rand}}, I_{\text{rand}})$ | 0.017 | 0.033 | 0.020 | 0.015 | 0.011 | 0.117 | -0.736 |
| $(I_{\text{shift}}, I_{\text{shift}})$ | 0.011 | 0.010 | 0.011 | 0.011 | 0.009 | 0.006 | -0.762 |
| $\mathcal{W}$ |  |  |  |  |  |  |  |
| $(\widehat{I}, \widehat{T})$ | 0.679 | 0.729 | 0.708 | 0.754 | 0.940 | 0.901 | / |
| $(I_{\text{center}}, T_{\text{center}})$ | 0.686 | 0.684 | 0.680 | 0.669 | 0.669 | 0.767 | -0.093 |
| $(\widetilde{I}, \widetilde{T})$ | 0.355 | 0.508 | 0.533 | 0.560 | 0.533 | 0.649 | -0.262 |
| $(I_{\text{rand}}, I_{\text{rand}})$ | 0.220 | 0.232 | 0.261 | 0.249 | 0.206 | 0.168 | -0.562 |
| $(I_{\text{shift}}, I_{\text{shift}})$ | 0.447 | 0.505 | 0.590 | 0.573 | 0.514 | 0.349 | -0.289 |
| recall@1 |  |  |  |  |  |  |  |
| $(\widehat{I}, \widehat{T})$ | 0.882 | 0.909 | 0.952 | 0.964 | 0.907 | 0.307 | / |
| $(I_{\text{center}}, T_{\text{center}})$ | 0.898 | 0.917 | 0.951 | 0.966 | 0.921 | 0.330 | +0.010 |
| $(\widetilde{I}, \widetilde{T})$ | 0.760 | 0.891 | 0.933 | 0.964 | 0.901 | 0.313 | -0.026 |
| $(I_{\text{rand}}, T_{\text{rand}})$ | 0.745 | 0.830 | 0.924 | 0.943 | 0.856 | 0.251 | -0.062 |
| $(I_{\text{shift}}, T_{\text{shift}})$ | 0.704 | 0.798 | 0.913 | 0.935 | 0.827 | 0.238 | -0.084 |

Table 9: Recall performance and modality gap in LAION's reconstructed embeddings, after the *shift* method and our concept based method to remove the gap, and after two baselines adapted to each method. As explained in Appendix I, the modality gap between the shift and concept based methods are not comparable, they can only be compared to their respective baselines. We can see that the shift method performs by far the worst in terms of recall. The concept based method seems to be consistently worse in terms of recall than it's baseline in terms of recall on smaller models, and equivalent in larger ones. However, in terms of modality gap, it is much better, with the mean baseline closing the gap in DiM but leaving it almost untouched in terms of distribution ($\mathcal{W}$).

| | CLIP | CLIP-L | OpenCLIP | OpenCLIP-L | SigLIP | SigLIP2 | $\Delta$ from $(\widehat{I}, \widehat{T})$ |
|---|---|---|---|---|---|---|---|
| **DiM** | | | | | | | |
| $(\widehat{I}, \widehat{T})$ | 0.812 | 0.808 | 0.714 | 0.747 | 1.043 | 1.051 | / |
| $(I_{\text{center}}, T_{\text{center}})$ | 0.354 | 0.379 | 0.472 | 0.469 | 0.375 | 0.507 | -0.437 |
| $(\widetilde{I}, \widetilde{T})$ | 0.153 | 0.186 | 0.192 | 0.216 | 0.123 | 0.100 | -0.701 |
| $(I_{\text{rand}}, T_{\text{rand}})$ | 0.292 | 0.311 | 0.366 | 0.360 | 0.308 | 0.420 | -0.520 |
| $(I_{\text{shift}}, T_{\text{shift}})$ | 0.354 | 0.379 | 0.471 | 0.467 | 0.374 | 0.507 | -0.437 |
| $\mathcal{W}$ | | | | | | | |
| $(\widehat{I}, \widehat{T})$ | 0.676 | 0.723 | 0.648 | 0.675 | 0.901 | 0.859 | / |
| $(I_{\text{center}}, T_{\text{center}})$ | 0.653 | 0.670 | 0.604 | 0.588 | 0.613 | 0.659 | -0.116 |
| $(\widetilde{I}, \widetilde{T})$ | 0.359 | 0.524 | 0.426 | 0.463 | 0.451 | 0.436 | -0.304 |
| $(I_{\text{rand}}, T_{\text{rand}})$ | 0.210 | 0.225 | 0.241 | 0.233 | 0.195 | 0.211 | -0.528 |
| $(I_{\text{shift}}, T_{\text{shift}})$ | 0.431 | 0.487 | 0.520 | 0.523 | 0.470 | 0.460 | -0.265 |
| **recall@1** | | | | | | | |
| $(\widehat{I}, \widehat{T})$ | 0.534 | 0.566 | 0.617 | 0.678 | 0.636 | 0.636 | / |
| $(I_{\text{center}}, T_{\text{center}})$ | 0.551 | 0.585 | 0.612 | 0.667 | 0.650 | 0.628 | +0.004 |
| $(\widetilde{I}, \widetilde{T})$ | 0.430 | 0.536 | 0.575 | 0.633 | 0.581 | 0.600 | -0.052 |
| $(I_{\text{rand}}, T_{\text{rand}})$ | 0.442 | 0.496 | 0.557 | 0.617 | 0.552 | 0.410 | -0.099 |
| $(I_{\text{shift}}, T_{\text{shift}})$ | 0.420 | 0.485 | 0.554 | 0.603 | 0.542 | 0.395 | -0.111 |

Table 10: Recall performance and modality gap in COCO's reconstructed embeddings, after the *shift* method and our concept based method to remove the gap, and after two baselines adapted to each method. Compared to Table 9, this table tells us that our concept based method transfers much better than all other ones to a different data distribution. Indeed, the performance in recall are comparable with those on LAION, yet our concept based method is the only one that seems to transfer it's ability to close the modality gap.

| | CLIP | CLIP-L | OpenCLIP | OpenCLIP-L | SigLIP | SigLIP2 | $\Delta$ from $(\widehat{I}, \widehat{T})$ |
|---|---|---|---|---|---|---|---|
| **DiM** | | | | | | | |
| $(\widehat{I}, \widehat{T})$ | 0.859 | 0.864 | 0.769 | 0.837 | 1.130 | 1.128 | / |
| $(I_{\text{center}}, T_{\text{center}})$ | 0.339 | 0.380 | 0.443 | 0.458 | 0.373 | 0.380 | -0.536 |
| $(\widetilde{I}, \widetilde{T})$ | 0.112 | 0.148 | 0.188 | 0.224 | 0.119 | 0.0881 | -0.785 |
| $(I_{\text{rand}}, I_{\text{rand}})$ | 0.288 | 0.317 | 0.361 | 0.459 | 0.321 | 0.340 | -0.583 |
| $(I_{\text{shift}}, I_{\text{shift}})$ | 0.340 | 0.381 | 0.443 | 0.365 | 0.373 | 0.381 | -0.551 |
| $\mathcal{W}$ | | | | | | | |
| $(\widehat{I}, \widehat{T})$ | 0.684 | 0.726 | 0.666 | 0.687 | 0.931 | 0.893 | / |
| $(I_{\text{center}}, T_{\text{center}})$ | 0.686 | 0.685 | 0.651 | 0.616 | 0.655 | 0.679 | -0.103 |
| $(\widetilde{I}, \widetilde{T})$ | 0.362 | 0.534 | 0.462 | 0.468 | 0.467 | 0.467 | -0.304 |
| $(I_{\text{rand}}, I_{\text{rand}})$ | 0.197 | 0.210 | 0.237 | 0.219 | 0.181 | 0.176 | -0.561 |
| $(I_{\text{shift}}, I_{\text{shift}})$ | 0.397 | 0.443 | 0.490 | 0.465 | 0.419 | 0.358 | -0.336 |
| **Zero shot accuracy** | | | | | | | |
| $(\widehat{I}, \widehat{T})$ | 0.350 | 0.472 | 0.400 | 0.504 | 0.466 | 0.448 | / |
| $(I_{\text{center}}, T_{\text{center}})$ | 0.367 | 0.484 | 0.406 | 0.522 | 0.490 | 0.490 | +0.020 |
| $(\widetilde{I}, \widetilde{T})$ | 0.340 | 0.477 | 0.401 | 0.524 | 0.481 | 0.460 | +0.007 |
| $(I_{\text{rand}}, I_{\text{rand}})$ | 0.310 | 0.438 | 0.398 | 0.511 | 0.481 | 0.409 | -0.016 |
| $(I_{\text{shift}}, I_{\text{shift}})$ | 0.311 | 0.434 | 0.387 | 0.497 | 0.479 | 0.381 | -0.025 |

Table 11: Zero-shot accuracy and modality gap in ImageNet's reconstructed embeddings, after the *shift* method and our concept based method to remove the gap, and after two baselines adapted to each method. Our conclusion here is the same as the conclusion for Table 10.

## J  DATASETS

Throughout this study, we used four datasets : LAION-400M (Schuhmann et al., 2021), COCO (Lin et al., 2014), ImageNet (Deng et al., 2009) and FashionIQ (Wu et al., 2021).

**LAION.**   We randomly selected a subset of 1M image-caption samples from the LAION-400M dataset. These samples were used to generate matching image-text embeddings. All reported SAEs were trained on this dataset.

**COCO.**   We used the train split of the 2017 update of the COCO dataset. Each image being paired with multiple captions, we generated pairs by duplicating the images for each of their captions. This dataset was used to further test the VLM encoders as well as the transferability of the learned features.

**ImageNet.**   This dataset was used to study the transferability of our features on a downstream classification task, commonly used in the literature. We generated about 80 captions for each class, embedded them, and averaged each class's embeddings to get a linear classifier, following Radford et al. (2021)'s method.

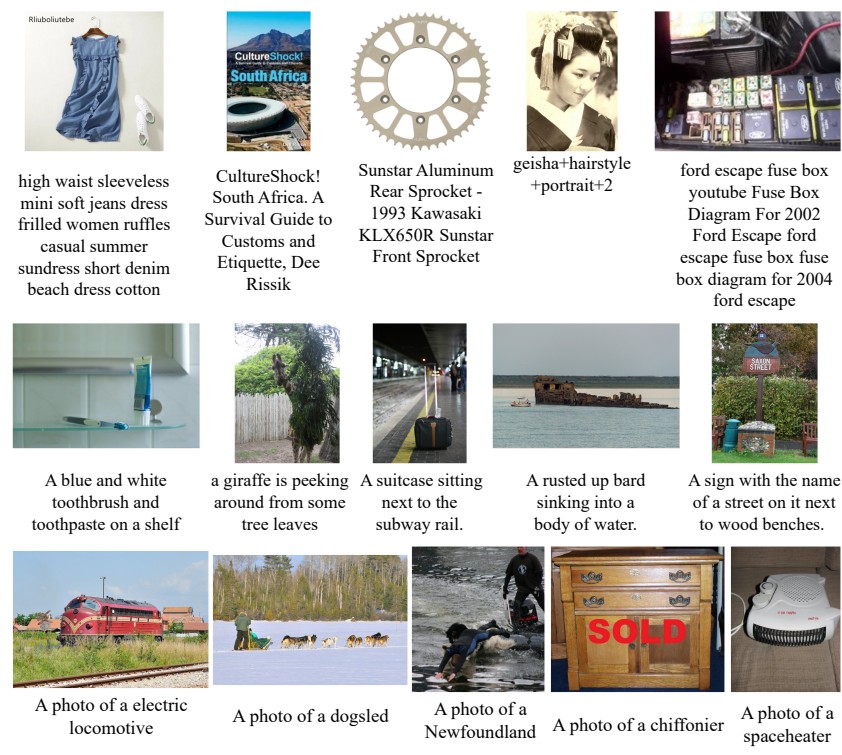

Figure 26: Examples of image-caption pairs in the LAION (**top**), COCO (**center**) and ImageNet (**bottom**).

**FashionIQ.**   This dataset is used to evaluate retrieval performance. It contains source images, called candidate, target images, and *relative* captions capturing the difference between the candidate and target. When several relative captions are available for a single candidate-target pair, we proceed as in ImageNet: we embed each caption and mean the results.

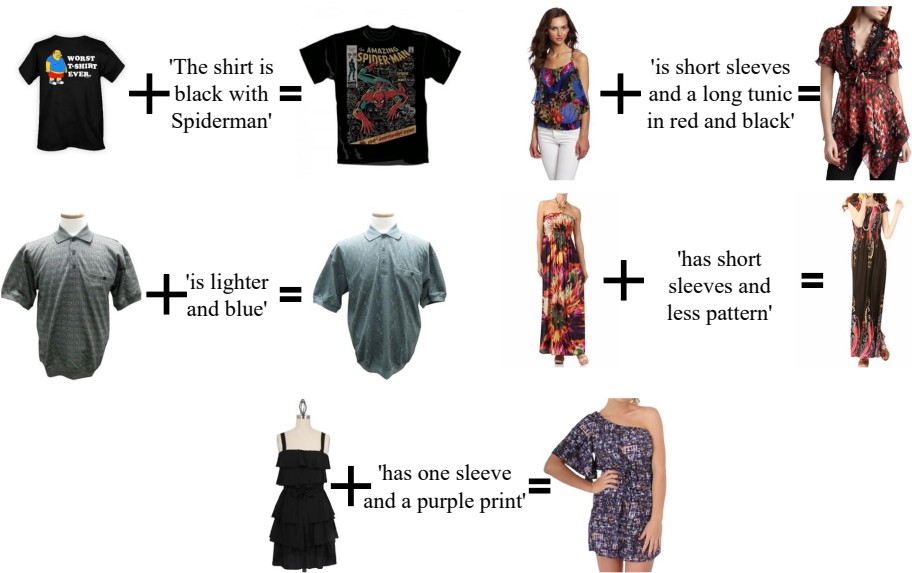

Figure 27: Examples of source and target images in FashionIQ, along with their relative caption.

## K  WHY REMOVE BIAS BEFORE COMPARING CONTENT

One could ask why it is important to remove unimodal information and what impact this has on retrieval and other downstream tasks. Can it help improve performance by cleaning the representation and revealing the useful semantic information? Building on the fact that many works have found that modality-specific components live in orthogonal subspaces, we are left with two choices: either the modality information acts as a constant offset, in which case removing or adding it does not change rankings, or the modality information is adaptive, as we find in this work. In the adaptive case, retrieval rankings can change significantly, making our decomposition a useful tool for precise control and steering of retrieval systems. We detail this here, and start with our setup.

**Setup**   We work in a representation space $\mathbb{R}^d$ that decomposes orthogonally as $\mathbb{R}^d = \Gamma \oplus \Omega$ into a content subspace $\Gamma$ and a modality subspace $\Omega$ with $\Gamma \perp \Omega$. For any underlying item $\boldsymbol{x}$, the observed embedding factorizes as

$$\boldsymbol{v} = \underbrace{\boldsymbol{\omega}(\boldsymbol{x})}_{\text{modality } \in \Omega} + \underbrace{\boldsymbol{\gamma}(\boldsymbol{x})}_{\text{content } \in \Gamma} \quad \text{with} \quad \mathbb{R}^d = \Gamma \oplus \Omega$$

where $\boldsymbol{\omega}(\boldsymbol{x}) \in \Omega$ captures modality-specific information and $\boldsymbol{\gamma}(\boldsymbol{x}) \in \Gamma$ represents the semantic content estimated by our SAE. The orthogonal direct sum decomposition ensures that content and modality components are geometrically independent.

For retrieval, we score against a set of unit-normalized candidates $\{\boldsymbol{y}_i\}_{i=1}^n$ using cosine similarity. To analyze pairwise rankings, we define the content margin $\Delta_c(i, j) := \langle \boldsymbol{\gamma}(\boldsymbol{x}), \boldsymbol{y}_i - \boldsymbol{y}_j \rangle$ and the modality margin $\Delta_m(i, j) := \langle \boldsymbol{\omega}(\boldsymbol{x}), \boldsymbol{y}_i - \boldsymbol{y}_j \rangle$. We also write $\alpha_i := \langle \boldsymbol{\omega}(\boldsymbol{x}), \boldsymbol{y}_i \rangle$ for the modality projection onto candidate $i$.

### K.1  CONSTANT MODALITY INFORMATION

We first consider the case where modality information affects all candidates equally. This corresponds to the intuitive scenario where modality acts as a simple offset that shifts all similarities by the same amount, leaving relative rankings unchanged.

When the modality component $\boldsymbol{\omega}(\boldsymbol{x})$ projects with identical strength onto every candidate in the retrieval set, removing this component preserves all pairwise orderings. This formalizes the intuition that a candidate-independent offset cannot affect relative rankings.

**Proposition 2** (Ranking invariance under constant projection).   *Assume unit normalized candidates* $\{\boldsymbol{y}_i\}_{i=1}^n$. *If there exists* $\alpha \in \mathbb{R}$ *such that* $\langle \boldsymbol{\omega}(\boldsymbol{x}), \boldsymbol{y}_i \rangle = \alpha$ *for all* $i$, *then for any* $i$ *and* $j$

$$\cos(\boldsymbol{v}, \boldsymbol{y}_i) > \cos(\boldsymbol{v}, \boldsymbol{y}_j) \quad \Longleftrightarrow \quad \cos(\boldsymbol{\gamma}(\boldsymbol{x}), \boldsymbol{y}_i) > \cos(\boldsymbol{\gamma}(\boldsymbol{x}), \boldsymbol{y}_j)$$

*The same equivalence holds with equality. The statement also holds for dot product scoring.*

*Proof.*   With $\|\boldsymbol{y}_i\| = 1$ we have

$$\cos(\boldsymbol{v}, \boldsymbol{y}_i) = \frac{\langle \boldsymbol{v}, \boldsymbol{y}_i \rangle}{\|\boldsymbol{v}\|} = \frac{\langle \boldsymbol{\omega}(\boldsymbol{x}), \boldsymbol{y}_i \rangle + \langle \boldsymbol{\gamma}(\boldsymbol{x}), \boldsymbol{y}_i \rangle}{\|\boldsymbol{v}\|}$$

Under the hypothesis $\langle \boldsymbol{\omega}(\boldsymbol{x}), \boldsymbol{y}_i \rangle = \alpha$ for all $i$ we get

$$\cos(\boldsymbol{v}, \boldsymbol{y}_i) - \cos(\boldsymbol{v}, \boldsymbol{y}_j) = \frac{\alpha - \alpha}{\|\boldsymbol{v}\|} + \frac{\langle \boldsymbol{\gamma}(\boldsymbol{x}), \boldsymbol{y}_i \rangle - \langle \boldsymbol{\gamma}(\boldsymbol{x}), \boldsymbol{y}_j \rangle}{\|\boldsymbol{v}\|} = \frac{\Delta_c(i, j)}{\|\boldsymbol{v}\|}$$

Hence $\cos(\boldsymbol{v}, \boldsymbol{y}_i) > \cos(\boldsymbol{v}, \boldsymbol{y}_j)$ if and only if $\Delta_c(i, j) > 0$. We also have

$$\cos(\boldsymbol{\gamma}(\boldsymbol{x}), \boldsymbol{y}_i) - \cos(\boldsymbol{\gamma}(\boldsymbol{x}), \boldsymbol{y}_j) = \frac{\Delta_c(i, j)}{\|\boldsymbol{\gamma}(\boldsymbol{x})\|}$$

The denominators $\|\boldsymbol{v}\|$ and $\|\boldsymbol{\gamma}(\boldsymbol{x})\|$ are positive constants independent of $i$ and $j$, so both differences have the same sign. If $\|\boldsymbol{\gamma}(\boldsymbol{x})\| = 0$ then $\boldsymbol{\gamma}(\boldsymbol{x}) = \boldsymbol{0}$ and all candidates tie under content while the first

difference is zero as well, so rankings are trivially invariant. The dot product case is identical without denominators. □

In the constant modality regime, removing modality information yields only the subspace of interest, and potentially more interpretable multimodal embeddings, but without any performance gains or losses. Thus, in that case, the decomposition provides the content representation by eliminating modality-specific artifacts, but since rankings remain identical, one should not expect improvements in retrieval metrics. This case represents a neutral scenario where the primary benefit is representational clarity rather than functional enhancement.

We now consider a near-constant case where modality projections vary slightly across candidates. This provides a useful practical criterion for when small variations cannot disrupt content-based rankings.

**Corollary 1** (Approximate invariance under bounded spread)**.** *Assume unit normalized candidates. Let $\varepsilon := \max_{i,j} |\alpha_i - \alpha_j|$. Then for any pair $(i, j)$ with $|\Delta_c(i,j)| > \varepsilon$, the inequality $\cos(\boldsymbol{v}, \boldsymbol{y}_i) > \cos(\boldsymbol{v}, \boldsymbol{y}_j)$ holds if and only if $\cos(\boldsymbol{\gamma}(\boldsymbol{x}), \boldsymbol{y}_i) > \cos(\boldsymbol{\gamma}(\boldsymbol{x}), \boldsymbol{y}_j)$.*

*Proof.* We have

$$\cos(\boldsymbol{v}, \boldsymbol{y}_i) - \cos(\boldsymbol{v}, \boldsymbol{y}_j) = \frac{\Delta_c(i,j) + \Delta_m(i,j)}{\|\boldsymbol{v}\|} \quad \text{with} \quad \Delta_m(i,j) = \alpha_i - \alpha_j$$

If $|\Delta_c(i,j)| > \varepsilon \geq |\Delta_m(i,j)|$ then $\Delta_c(i,j)$ and $\Delta_c(i,j) + \Delta_m(i,j)$ have the same sign. Division by the positive constant $\|\boldsymbol{v}\|$ preserves sign. The sign also matches the sign of $\Delta_c(i,j)$ which determines the sign of $\cos(\boldsymbol{\gamma}(\boldsymbol{x}), \boldsymbol{y}_i) - \cos(\boldsymbol{\gamma}(\boldsymbol{x}), \boldsymbol{y}_j)$. □

This corollary establishes a robustness condition: when modality variations are small relative to content differences—even in the presence of multiple adaptive modality components – the content signal prevails and rankings remain unchanged. Like the constant case, decomposition offers no performance gains here since content already drives the original rankings.

However, as we will show next and as we empirically find in this work, real modality information is typically adaptive rather than constant, leading to significant ranking changes when removed.

## K.2 Adaptive modality information

This is what we observe empirically in this work. The modality signal decomposes into multiple unimodal atoms and the modality projection varies significantly across candidates. In this regime, removing $\boldsymbol{\omega}(\boldsymbol{x})$ can dramatically change rankings. The next proposition provides a necessary and sufficient condition for ranking flips and clarifies that flips occur when the candidate-dependent modality margin opposes and dominates the content margin.

**Proposition 3** (Characterization of ranking flips under adaptive modality)**.** *Assume unit normalized candidates. For any pair $(i, j)$, the two differences*

$$\cos(\boldsymbol{v}, \boldsymbol{y}_i) - \cos(\boldsymbol{v}, \boldsymbol{y}_j) \quad and \quad \cos(\boldsymbol{\gamma}(\boldsymbol{x}), \boldsymbol{y}_i) - \cos(\boldsymbol{\gamma}(\boldsymbol{x}), \boldsymbol{y}_j)$$

*have opposite signs if and only if*

$$\Delta_c(i,j)\big(\Delta_c(i,j) + \Delta_m(i,j)\big) < 0$$

*Equivalently, the signs of $\Delta_m(i,j)$ and $\Delta_c(i,j)$ are opposite and $|\Delta_m(i,j)| > |\Delta_c(i,j)|$. The same characterization holds for dot product scoring.*

*Proof.* Using the identities above we write

$$\cos(\boldsymbol{v}, \boldsymbol{y}_i) - \cos(\boldsymbol{v}, \boldsymbol{y}_j) = \frac{\Delta_c(i,j) + \Delta_m(i,j)}{\|\boldsymbol{v}\|} \quad \text{and} \quad \cos(\boldsymbol{\gamma}(\boldsymbol{x}), \boldsymbol{y}_i) - \cos(\boldsymbol{\gamma}(\boldsymbol{x}), \boldsymbol{y}_j) = \frac{\Delta_c(i,j)}{\|\boldsymbol{\gamma}(\boldsymbol{x})\|}$$

The denominators are positive constants independent of the candidates. Therefore the two differences have opposite signs if and only if $\Delta_c(i,j)$ and $\Delta_c(i,j) + \Delta_m(i,j)$ have opposite signs. This is

equivalent to the strict inequality $\Delta_c(i,j)\big(\Delta_c(i,j)+\Delta_m(i,j)\big) < 0$. Expanding gives the equivalent condition that $\Delta_m(i,j)$ and $\Delta_c(i,j)$ have opposite signs and that the magnitude of $\Delta_m(i,j)$ exceeds the magnitude of $\Delta_c(i,j)$. The dot product case follows by the same steps with no denominators. $\quad\square$

In practice, visual information or bias is almost orthogonal to textual information or bias. This makes $\Delta_m(i,j) \approx 0$, and therefore provides a guarantee that rankings should stay the same under sole manipulation of modality-specific information.

**Remark 1** (Concrete two-dimensional example). *Let $\Omega = \mathrm{span}\{e_1\}$ and $\Gamma = \mathrm{span}\{e_2\}$ with $e_1$ orthogonal to $e_2$. Take $\boldsymbol{\omega}(\boldsymbol{x}) = \boldsymbol{e}_1$ and $\boldsymbol{\gamma}(\boldsymbol{x}) = \frac{1}{2}\boldsymbol{e}_2$. Choose candidates $\boldsymbol{y}_1 = \frac{1}{\sqrt{2}}(\boldsymbol{e}_1 + \boldsymbol{e}_2)$ and $\boldsymbol{y}_2 = \frac{1}{\sqrt{2}}(-\boldsymbol{e}_1 + \boldsymbol{e}_2)$ which are unit normalized. Then $\Delta_c(1,2) = 0$ and $\Delta_m(1,2) = \sqrt{2}$ so $\cos(\boldsymbol{v}, \boldsymbol{y}_1) > \cos(\boldsymbol{v}, \boldsymbol{y}_2)$ while $\cos(\boldsymbol{\gamma}(\boldsymbol{x}), \boldsymbol{y}_1) = \cos(\boldsymbol{\gamma}(\boldsymbol{x}), \boldsymbol{y}_2)$. If we perturb $\boldsymbol{\gamma}(\boldsymbol{x})$ slightly toward $-\boldsymbol{e}_2$ so that $\Delta_c(1,2) < 0$, the inequality reverses under content while it stays the same under observed scores, demonstrating a strict ranking flip.*

In summary, if modality information were candidate-independent, removing it would leave retrieval unchanged as shown in Proposition 2. In practice, however, modality information is multi-component and adaptive. Its projections onto candidates differ significantly, and when those candidate-dependent differences oppose and dominate the content margin, rankings change dramatically as characterized in Proposition 3. This is precisely why we identify and remove unimodal atoms before comparing content – doing so can reveal the true content-based similarities that were previously masked by conflicting modality signals. The bounded spread criterion in Corollary 1 provides a practical stability guarantee, ensuring that decomposition is beneficial primarily when modality variations are large relative to content margins.

## L   On previous work's attempt to characterise $\Omega_I \oplus \Omega_T$

We compare the method to remove the modality gap proposed by Schrodi et al. (2025) to ours and to three baselines in its capacity to close the modality gap. We report measures the modality gap as (*i*) $\|\Delta\|$ the norm of the difference in means, (*ii*) sep the accuracy score of a linear logistic regression probe trained to classify text vs image embeddings, measuring the linear separability of both distribution, (*iii*) RMG, a metric introduced by Schrodi et al. (2025) and (*iv*) our OOD score.

Let $I, T \in \mathbb{R}^{N \times d}$ be image-caption pairs embedded using CLIP. Let $\mu_I := \mathbb{E}[I_i]$, $\mu_T := \mathbb{E}[T_i]$ and $\Delta := \mu_T - \mu_T$. Let $e_1, ..., e_k$, $k \ll d$ be the directions identified by Schrodi et al. (2025).

**Baseline A:** center embeddings modality-wise, $I \leftarrow I - \mu_I$, similarly for $T$. **Baseline B:** project out modality wise means, $I \leftarrow I - (I \cdot \mu_I)\mu_I$, with $\mu_I$ being additionally normalised, and similarly for $T$. **Baseline C:** project out $\Delta$, $I \leftarrow I - (I \cdot \Delta)\Delta$, with $\Delta$ being normalized, and similarly for $T$. **Method:** project $I$ and $T$ on the orthogonal complement of $\mathrm{Span}(e_1, ..., e_k)$. We also compare to our method and to the original embedding (line "unchanged" in Table 12).

| Method | $\|\Delta\|$ ($\downarrow$) | sep ($\downarrow$) | RMG ($\downarrow$) | OOD ($\downarrow$) |
|---|---|---|---|---|
| Unchanged | 0.72 | 1.00 | 0.52 | 1.00 |
| Baseline A | 0.00 | 0.50 | 0.45 | 0.99 |
| Baseline B | 0.00 | 0.50 | 0.45 | 0.98 |
| Baseline C | 0.00 | 0.50 | 0.45 | 0.89 |
| Schrodi et al. | 0.22 | 1.00 | 0.46 | 0.98 |
| Ours | 0.01 | 0.50 | 0.41 | 0.63 |

Table 12: Comparison of 2 methods and 3 baselines in their ability to remove the modality gap, as measured by 4 metrics.