# OpenReview forum: "Cross-Modal Redundancy and the Geometry of Vision–Language Embeddings"
_ICLR.cc/2026/Conference — ICLR 2026 Poster_

### Official Review · Reviewer_XysU · 2025-10-19

**Soundness:** 3
**Presentation:** 2
**Contribution:** 2
**Rating:** 2
**Confidence:** 5

**Summary:**

The paper investigates the geometry of the embedding space of CLIP-like models using sparse autoencoders. The authors augment SAE training by an iso-energy regularization term that encourages SAE latents to have similar spreads (i.e., second moments) for both modalities (Def. 2). They show that only a small set of features explain the modality gap and the remaining features are sufficient for cross-modal alignment. Removing former features reduces modality gap while retaining performance and the latter allows for vector arithmetic.

**Strengths:**

* S1: The energy penalty (Def. 2/ Eq. 1) is interesting and a simple addition to the SAE loss.

* S2: Synthetic & real experiments confirm that the proposed aligned SAE better matches the geometry of CLIP-like models. Particularly, if the SAE features can distinguish between shared or modality-gap-specific.

* S3: The paper introduced four metrics to evaluate whether the SAE variants capture the geometrical or functional properties of the VLMs.

* S4: The proposed SAE allows semantic vector arithmetic.

**Weaknesses:**

* W1: 3 out of the 4 key findings have been reported in previous work (see bullet points below). While the findings are reached using a different, more complex approach, the current paper seems to re-report these findings.
	* Few (unimodal) features fully explain the modality gap (Fig. 2 left, 3) ~> see Fig. 4 in [3] or Fig. 3 in [4]
	* Bimodal features carry the entire cross-modal alignment signal (Fig. 2 right, Fig. 3) ~> cross-modality transferability experiments in [2], e.g., Tab. 2.
	* Removing those modality-gap features reduces the modality gap without loss of performance (Fig. 4) ~> again, see cross-modality experiments in [2], e.g., Tab. 2.

* W2: The paper provides little to no experimental details in the main text, making it hard to understand the results without searching the supplemental.

* W3: It is assumed that bimodal atoms are semantically aligned across modalities (“bimodal atoms encode the shared conceptual backbone” l. 345) and few qualitative examples are provided in Appendix G. However, there is no quantitative evaluation for this claim.

* W4: Only contrastive models are evaluated. For example, the modality gap has been also observed in multimodal LLMs. It’d be important to include such results.

## Comment

* C1: I’d encourage the authors to include discussions on missing relevant literature [1-4].

* C2: This work’s proposition 1 seems closely related to [2]’s proposition A.1. The only difference seems to be that the modality information can be adaptive here.

* C3: The caption of Fig. 4 is partially occluded from Fig. 5.

---

[1] https://www.mlmi.eng.cam.ac.uk/files/2021-2022_dissertations/understanding_and_fixing_the_modality_gap_in_vision-language_models_reduced.pdf

[2] https://openreview.net/forum?id=D-zfUK7BR6c

[3] https://openreview.net/forum?id=uAFHCZRmXk

[4] https://openreview.net/forum?id=QGUju9B68Z

**Questions:**

* Q1: Is the standard SAE (l. 176/177, 185) the MP-SAE or is it truly standard SAE?

* Q2: How are unimodal or bimodal features separated?

* Q3: Do the unimodal features approximate the modality gap vector? Related to that, does it explain why they all have such high cosine similarities (Fig. 16b)?

* Q4: What is $\mu$ in Fig. 2 left?

---

> ### Author Response · Authors · 2025-11-19
> **Response - Part I**
>
> We thank the reviewer for their very careful and constructive assessment. They raise fair points regarding previous work, and we acknowledge that several phenomena we discuss have indeed been observed before in related contexts.
>
> We want to be very clear regarding your concern that we might be re-reporting known findings: **we are well aware of the foundational work you cited, particularly regarding the existence of the modality gap and the general transferability of representations and we certainly do not claim to have discovered these phenomena.**
>
> Now concerning our precise contributions. While previous work described the modality gap geometrically (as a cone or a mean shift) or dynamically (as a result of contrastive loss), our work identifies **the specific "atomic" units that constitute it**. We show that the gap is not just a statistical artifact but is physically encoded by a distinct set dictionary atoms. **This shift from a macro-level geometric description to a micro-level concept decomposition allows for more precise interventions**. For instance, regarding the "few atoms" observation: we find that while a few high-energy atoms explain the majority of the variance (the bias), removing the entire unimodal subspace (which is complex and high-dimensional) is necessary to fully close the gap without performance loss. **This distinction between the "bias" (mean shift) and the "subspace" (manifold structure) is a nuance that concept-level analysis reveals, which static geometric baselines often miss.**
>
> Note : all references to our article are with respect to the revised pdf.
>
> ---
>
> **Clarifications on our contributions and links to prior work:**
>
> We respectfully disagree with the reviewer, we are well aware of previous work and extended the related work in a new section dedicated to contextualize our contributions more clearly [Section 2 of the revised pdf].
> To recall, **our novelty lies not in the description of the phenomena discussed in the new related work section, but rather in the characterization of these structures in foundational VLM encoders**. Through the use of a concept-based approach, we are able to identify (*i*) high energy unimodal features to unimodal biases, (*ii*) $\Gamma = \mathrm{Cone}(\delta \cdot D)$ and $\Omega_{I/T} = \mathrm{Cone}(\delta_{I/T}\cdot D)$. Here, $\mathrm{Cone}(e_1...ek)=${$\sum_{i=1}^k\lambda_ie_i \mid \lambda_i \geq 0$}, $D \in \mathbb{R}^{K\times d}$ is the set of $K$ dictionary atoms of latent dimension $d$, and $\delta$ (resp. $\delta_I$, $\delta_T$) $\in (0,1)^d$ is the binary mask selecting bimodal (resp. image-, text-only) atoms. To claim this characterization, we carefully analyze four complementary aspects of the dictionary through novel metrics. Once validated, we further test its practical value through targeted interventions on these structures.
>
> **We do claim** the discovery of the geometric phenomena that foundational VLM encoders organise their conceptual structure into linear subspaces $\Gamma$ and $\Omega$.
> - [10] shows that theoretical models should have some modality specific variations of their representation to preserve unimodal capabilities and account for modality specific information, which (i) might not be the case in contrastive models, for they are purely trained on a cross modal alignment objective, (ii) might not be organised through neatly structured linear information-subspaces, (iii) they do not show happens in unregularized (or regularized) VLM encoders but use as a motivation to introduce a new architecture improving upon the contrastive framework by explicitly adding regularisation to favorise the emergence and the structuration of such modality specific variations.
> - [3]'s findings align with [10]'s in showing that modality specific information shapes the representations. **Their result do not show, nor do they claim to show, that $\Omega = Span(e_1, ..., e_k)$**. They discover that the difference in means has disproportionately high dot product with few canonical directions, and use this surprising observation to ask whether they can use it to close the gap. With that said, we can indeed extrapolate over their discussion of Section 4.2. to say that they discover that $\Omega = Span(e_1, ..., e_k)$ with $e_i$ canonical basis vectors and $k \ll d$. However, their analysis of the validity of such a characterization is lacking, showing clear post-intervention performance loss in cross-modal tasks (see their Figure 4) which **indicates a significant alteration of $\Gamma$ in trying to isolate $\Omega$, whilst leaving the gap wide open** (see Appendix L).
> - To the best of our knowledge, we are the first to explicitly make such a description, and successfully characterise it in real-world foundational VLMs.
>
> **We would further like to emphasize the significance of our Proposition 1** compared to the weaker version in [2], **especially in the context where we show the existence of such subspaces.** [... continuing in Part II]

---

> ### Author Response · Authors · 2025-11-19
> **Response - Part II**
>
> [...] It shows that adaptive, multi dimensional modality specific information has non trivial effects on downstream performance, except if it is explicitly contained in a subspace orthogonal to that of shared, cross modal information.
>
> -----
>
> **Specific Replies to the Main Weaknesses**
>
> *W1 — Similarity to previous findings*
>
> Now that the above context is laid in light of the new related work section, we address the specific points duly reported.
> 1) the “few unimodal atoms” you refer to could be one of the two following. (i) The “few high energy atoms” which we find to be truly just variations of a single one (an artifact of training using the MP procedure) corresponding to unimodal biases - in this case, **they do *not* fully explain the modality gap, as we show that the whole unimodal subspace is required for that [figure 4]**. They do explain the overwhelming majority of energy contained in unimodal atoms. (ii) All unimodal atoms, spanning $\Omega_I\oplus\Omega_T$, which are not few but similar in number to bimodal atoms. **This suggests that $\Omega$ is much more complex than a simple mean or canonical directions where these means are salient**.
> 2) **We do not claim to discover cross modal transferability. In fact, this is one of the motivations for our work.** We mention VLM’s success in cross modal tasks : “Vision–language models (VLMs) align images and text with remarkable success,” - l11, “creating shared embedding spaces where visual and textual representations of similar concepts align. Despite their empirical success,” - l34,35. Yet, despite this success, little is known on *how* this alignment, and subsequent transfer capabilities, occurs “, we lack a principled understanding of how these models internally organize and align semantic content across modalities” - l35. The shared vs specific information subspaces $\Gamma$ and $\Omega$, suggested by [10] gives a hint to answer that question, yet many possible internal organisations of $\Gamma$ with the same apparent capabilities are possible (cf. apparent idiosyncratic atoms found by [11] and shared ones found by us). **Our claim is therefore not that we discover transferability, but that we discover how it is managed at the concept level.**
> 3) cf discussion above. Indeed, the similarity between [2]’s proposition A.1. and our proposition 1 (cf. comment C2, discussion on that below) explains this similarity in performance preservation. However, the approach is very much different: **they consider modality wise means, which, as we showed, is not enough to close the gap, while we consider the whole modality specific subspace $\Omega$.**
>
> *W2 — Experimental details*
>
> This is true and we apologize for that. The length of the work, with over 40 pages in total, required us to place a substantial amount of material in the appendix which limited how much experimental detail could fit in the main text. We appreciate the reviewer’s deep analysis despite this unusual volume of work. For the camera-ready version, we will bring key elements from the appendix into the main body to improve clarity and make the results easier to follow.
>
>
> *W3 — Quantitative evaluation of bimodal semantic consistency*
>
> We are not sure to understand the point of the reviewer.
> The specific quote you mention refers to the following result : “$\Gamma = \mathrm{Cone}(\delta \cdot D)$”.
> As for the semantic stability of bimodal vs unimodal atoms across both modalities, **we do have quantitative results about that** : this is precisely what is summarized in our novel quantitative metrics $\rho$ and FDA. In figure 12, the gray dashed line in the purple and yellow regions indicate the proportion of “self-bridge” B_i_i (below the line) to the proportion of “cross-bridge” involving two different atoms B_i_j (above the line). Keeping in mind that the number of cross-bridge entries is square that of self-bridge entries, this tells us that **bimodal atoms are extremely stable across both modalities while unimodal ones are not.** In the case of our aligned-SAE, the latter do not bridge at all to the other modality – we compared to null models with randomly shuffled values.
>
> *W4 — Evaluation on multimodal LLMs*
>
> Fair point, we agree with the reviewer. As our method only requires a shared embedding, it can in principle extend beyond contrastive models. We mention this as a proposed direction for future work in the discussion (l462-464), and are currently working on extensions of our work on such models.

---

> ### Author Response · Authors · 2025-11-19
> **Response - Part III**
>
> *C2 (Prop. 1)* — That seems to be indeed very relevant, thanks for pointing that out! We include a note on this previous result in the relevant sections (l143 and l414). The difference is indeed that they consider a constant, the difference in means, where we consider the whole modality specific subspace $\Omega_I \oplus \Omega_T$.
>
> *C3 (Figure 4 caption)* — The occlusion issue will be fixed in the revised PDF.
>
> *Q1. Is the “standard SAE” the MP-SAE or a ReLU/Top-K SAE?*
> It is the unregularized Matching Pursuit (MP) SAE. For brevity we refer to it as “SAE”; “SAE-A” denotes the MP-SAE with the alignment term (Eq. 1). We chose MP for its sparse generative interpretation and strong reconstruction performance (Appendix A). The main observations (bimodal atoms carrying alignment, unimodal atoms explaining the gap) are robust across SAE variants.
>
> *Q2. How are unimodal and bimodal atoms separated?*
>
> We compute the “modality score” $\mu$ for each atom, which has a value between 0 and 1. If $\epsilon < \mu < 1-\epsilon$, the atom is labeled as bimodal. Otherwise, it is unimodal. We detail how we compute $\mu$ and calibrate $\epsilon$ in Appendix E.2.2. paragraphs “Modality Score.” and “Thresholding μ.” We add a note on this point (l105) as it is essential and indeed lacking from the main body. Thanks for pointing that out!
>
> *Q3. Do unimodal features approximate the modality-gap vector, explaining their high mutual cosine?*
>
> Great question! With reconstructed embeddings Â = ZD, the mean difference (gap) between modalities is
> g = μ_img − μ_txt ≈ (E[Z | img] − E[Z | txt]) D.
> Because bimodal atoms have near-equal second moments, their contribution cancels, leaving g concentrated in the span of unimodal atoms. This partially explains unimodal atom’s alignment with modality-specific cones (as reflected by pacc). However, **it does not explain the high cosine similarity observed in high energy unimodal atoms** ! This is actually an artifact of the training dynamics of an MP Sparse Autoencoder. It essentially learns duplicates of the unimodal biases. Cosine similarity between this bias and arbitrary low energy unimodal feature is much lower, closer to 0. As the mutual cosine similarity between arbitrary low energy unimodal atoms is close to 0, their contribution to the above decomposition of D cancels each other.
>
> *Q4. What is $\mu$ in Fig. 2 (left)?*
> $\mu$ is the modality score mentioned above. The plot shows per-atom energies E_k^img (x-axis) versus E_k^txt (y-axis) - in log space. Thus, points of equal modality score appear in 45° lines. We fix this lack of label in the revised pdf.
>
> -----
>
> **Revisions to PDF**
>
> - We include a new related work section entirely dedicated to clarifying existing descriptions of multimodal phenomena and the link with our contributions.
> - To address C2, we add a mention to [2]’s Proposition 1 which indeed precedes our own (l143 and l414)
> - Minor clarifications and corrections mentioned above.
>
> -----
>
>
> **Closing Remarks**
>
> Thank you again for your time and for holding our work to a high standard. We believe that by explicitly positioning our findings as a "concept-level mechanism" for known geometric phenomena, rather than a discovery of the phenomena themselves, we can resolve the concerns regarding novelty.
>
> ----
>
> [10] Jiang et al. (2023) "*Understanding and Constructing Latent Modality Structures in Multi-Modal Representation Learning*"
>
> [11] Papadimitriou et al. (2025) "*Interpreting the linear structure of vision-language model embedding spaces*"

---

### Official Review · Reviewer_BMn2 · 2025-10-30

**Soundness:** 3
**Presentation:** 3
**Contribution:** 3
**Rating:** 6
**Confidence:** 3

**Summary:**

This paper studies the geometry of VLM embedding spaces via the Iso‑Energy Assumption—shared concepts should have domain‑invariant average squared activation. The authors train an Aligned Matching‑Pursuit SAE with a small cross‑modal alignment regularizer, yielding a dictionary that separates bimodal atoms (which carry all cross‑modal alignment) from unimodal atoms (a few high‑energy modality‑specific biases explaining the modality gap). On synthetic data and CLIP/OpenCLIP/SigLIP variants, this preserves reconstruction while markedly improving multimodality metrics, and enables interventions such as removing unimodal atoms to close the modality gap without hurting retrieval and performing in‑distribution semantic arithmetic restricted to the bimodal subspace.

**Strengths:**

1.  **Clear Problem Formulation and Strong Motivation:** The paper articulates a pertinent and significant problem in VLM interpretability and manipulability. By focusing on the geometric underpinnings of cross-modal alignment and the "modality gap," the work addresses a critical area for improving VLM transparency and control.
2.  **Novel and Intuitive Hypothesis:** The "Iso-Energy Hypothesis" offers an elegant and interpretable statistical prior for identifying shared concepts within a sparse dictionary. This hypothesis provides a concrete, measurable criterion that transforms the abstract notion of "cross-modal redundancy" into an actionable constraint for dictionary learning.
3.  **Demonstrated Practical Interventions:** The ability to close the modality gap by masking uni-modal atoms and to perform "in-distribution" semantic arithmetic within the bi-modal subspace represents a significant practical contribution. These interventions offer concrete pathways for improving the robustness and interpretability of VLM applications.

**Weaknesses:**

1.  **Reliance on Paired Data for Alignment Regularization:** Although lines 158-160 allude to the potential of leveraging "cross-modal redundancy alone," the current formulation of the alignment regularizer explicitly requires instance-level image-text pairs. The robustness of the method to noisy or imperfect pairings, or its applicability in settings with weak or no explicit pairings (e.g., using only domain labels), remains unexplored. This dependency may limit its generality and practical scope.
2.  **Limited Assessment of Dictionary Stability and Generalizability:** While the paper aims to enhance SAE dictionary stability via the Iso-Energy Assumption and demonstrates improved recovery on synthetic data during "Sanity check", it lacks a systematic and multi-faceted analysis of this robustness on large-scale real-world VLM datasets. The reproducibility of the learned dictionary under varying conditions, such as different expansion ratios, sparsity targets, or subsets of training data, remains unexplored. Thus, the evaluation of this crucial aspect in practical scenarios is not yet comprehensive.
3.  **Scope of Evaluation and Downstream Task Relevance:** While the paper demonstrates strong results on retrieval-oriented metrics and interventions, the generalizability to other VLM tasks (e.g., visual question answering, image generation, localization, counting, spatial reasoning) is not fully explored. The claim that "masking uni-modal atoms does not hurt performance" might hold for certain tasks, but could be detrimental for tasks that rely on more modality-specific information.

**Questions:**

**External Validation of Atomic Concepts:** While visualizations are provided, the "semantic stability" of the atoms is largely qualitative. Is it possible to introduce quantitative measures for concept purity, namability, or alignment with human annotations to further validate the interpretability and meaningfulness of the identified bi-modal and uni-modal atoms? This would provide stronger evidence that the method is indeed recovering genuine, human-understandable concepts.

---

> ### Author Response · Authors · 2025-11-19
> **Response - Part I**
>
> Thank you for your detailed and constructive review! We appreciate the recognition of the novelty of our hypothesis-driven approach and its practical implications.
>
> -----
>
> > _"Reliance on Paired Data for Alignment Regularization [...] the current formulation of the alignment regularizer explicitly requires instance-level image-text pairs [...] This dependency may limit its generality and practical scope."_
>
> Excellent point. We tried an L1 and an L2 formulation of the same loss. If formulated at the instance level (requiring pairs), results are indistinguishable. However, if formulated at the population level, the performance of the dictionary significantly degrades. **When taking the exact formulation of Definition 2, the expectation is approximated on each batch leading to poor statistics** as features are sparse and have low frequency – typically much smaller than the inverse of the batch size. Solving this issue is critical for generalising our methods to other architectures.
>
>
> > _"Limited Assessment of Dictionary Stability and Generalizability [...] The reproducibility of the learned dictionary under varying conditions, such as different expansion ratios, sparsity targets, or subsets of training data, remains unexplored. Thus, the evaluation of this crucial aspect in practical scenarios is not yet comprehensive."_
>
>  - (1) It is true that our stability results focus on SAEs trained under identical hyperparameters and that the SAE literature could benefit from larger-scale stability experiments. This goes beyond the scope of our study. **We do report the study of many SAE architectures, one set of which are small-scale [table 3], and another set of large-scale SAEs [table 4]**. Results on all metrics are consistent across both scale and architecture. For the stability metrics in particular, they are quite hard to interpret, but we report them nonetheless.
>  - (2) **We study the generalizability of our dictionaries to completely different data distributions and tasks**: all SAEs reported were trained on a subset of 1M pairs from the LAION400M dataset and evaluated on (i) a test set of the same data distribution, as well as on (ii) COCO image-caption pairs, (iii) ImageNet with hand crafted class wise prompts following [1], and (iv) FashionIQ triplets of source/target image and textual description of the difference [tables 8-11].
>
>
> > _"Scope of Evaluation and Downstream Task Relevance [...] The claim that "masking uni-modal atoms does not hurt performance" might hold for certain tasks, but could be detrimental for tasks that rely on more modality-specific information."_
>
> That is very much true. **Removing modality-specific information will hurt performance if the task relies on this information**. Our claim is that we have proof that ranking-based tasks are unaffected, and that this allows us to remove the gap where previous work fails. But removing the gap might not be desirable if the task needs the information it contains. **The actual takeaway is not the fact that we remove the gap, but that we are actually able to do so with enough precision to keep all predictive shared information in place**, which provides evidence we have successfully isolated shared information, **allowing for more targeted/grounded/interpretable interventions**. Future work could explore how various types of tasks rely on information contained in the shared vs specific subspaces we elicit.

---

> ### Author Response · Authors · 2025-11-19
> **Response - Part 2**
>
> > _"External Validation of Atomic Concepts: While visualizations are provided, the "semantic stability" of the atoms is largely qualitative. Is it possible to introduce quantitative measures for concept purity, namability, or alignment with human annotations [...] ? [...]"_
>
>  - (1) **The question of alignment with human interpretability is indeed very interesting, especially in the context of multimodality**. This goes beyond the scope of our study but would definitely be an interesting direction for future work.
>  - (2) As for the semantic stability of bimodal vs unimodal atoms across both modalities, **we do have quantitative results about that** : this is precisely what is summarised in our novel quantitative metrics $\rho$ and FDA. In figure 12, the gray dashed line in the purple and yellow regions indicate the proportion of “self-bridge” B_i_i (below the line) to the proportion of “cross-bridge” involving two different atoms B_i_j (above the line). Keeping in mind that the number of cross-bridge entries is square that of self-bridge entries, this tells us that **bimodal atoms are extremely stable across both modalities while unimodal ones are not**. In the case of our aligned-SAE, the latter do not bridge at all to the other modality - we compared to null models with randomly shuffled values.
>
> -----
>
> [1] Radford et al. (2021) "Learning Transferable Visual Models From Natural Language Supervision"

---

### Official Review · Reviewer_Yfxj · 2025-10-31

**Soundness:** 3
**Presentation:** 2
**Contribution:** 2
**Rating:** 4
**Confidence:** 3

**Summary:**

The paper studies the geometry of vision–language embeddings through a proposed Iso-Energy assumption, which states that shared cross-modal concepts should have equal activation energy across modalities. To explore this, the authors introduce an aligned sparse autoencoder (SAE-A) that adds a cosine-similarity–based alignment loss to a standard sparse autoencoder. The numerical experiments on CLIP, OpenCLIP, and SigLIP embeddings show that the aligned SAE could improve cross-modal alignment metrics while maintaining reconstruction quality.

**Strengths:**

1. The paper provides an interesting perspective on the geometry of vision–language embeddings by introducing the Iso-Energy assumption.

2. The numerical results are consistent, showing that the aligned SAE can improve cross-modal alignment metrics without damaging reconstruction quality.

**Weaknesses:**

1. The connection between the Iso-Energy Assumption in Definition 2 and the implemented loss in Equation (1) is not that clear. Definition 2 describes a population-level equality of per-coordinate activation energies across modalities, whereas the alignment loss in (1) simply quantifies the batch-level sum of cosine similarity between sample codes. The paper does not provide a derivation or justification showing that this cosine similarity sum term directly enforces or meaningfully approximates the Iso-Energy property.

2. The alignment loss in Equation (1) effectively reduces to a vanilla sum of cosine similarities between the latent codes from two modalities. This formulation looks too simple and somewhat ad hoc, lacking a clear connection to encourage equalized energy statistics as defined by the Iso-Energy assumption.

3. The paper introduces the aligned sparse autoencoder without providing sufficient background on the baseline SAE formulation, its reconstruction, and sparsity terms. This makes the method less self-contained and more difficult for readers less familiar with the SAE framework to follow.

4. Some of the mathematical definitions, particularly in Definition 2, are not presented rigorously. The conditional expectation is written as if conditioned on the specific sample $X$, which collapses the expectation to the outcome for that given value of $X$ in the conditional expectation of (1).

**Questions:**

1. Can the authors clarify the precise theoretical link between the Iso-Energy Assumption in Definition 2 and the cosine-similarity–based alignment loss in Equation (1)?

2. As the alignment loss in (1) reduces to a simple sum of cosine similarities, did the authors experiment with other similar regularizers (e.g., the sum of the squared or absolute value of the inner products in (1)) or other regularizers that can more directly enforce the Iso-Energy property in Definition 2?

---

> ### Author Response · Authors · 2025-11-19
> **Response**
>
> **We thank the Reviewer for the careful reading** and for raising interesting questions ! We appreciate the opportunity to clarify some theoretical aspect and our design choices.
>
> ---
>
> > _"The connection between the Iso-Energy Assumption [...] and the implemented loss in Equation (1) is not that clear. [...] The paper does not provide a derivation or justification showing that this cosine similarity sum term directly enforces or meaningfully approximates the Iso-Energy property."_
>
> This is a crucial point. We clarify that the link relies on the specific normalization constraints of our implementation, which we will make more explicit in the revision.
>
> As noted in Section 2 (Line 316), we operate on $l_2$-normalized codes $Z^{(d)}$. **Thus, for unit-norm vectors $Z^{(d)}$ and $Z^{(d')}$, maximizing their cosine similarity (Equation 1) is equivalent to minimizing their Euclidean distance $||Z^{(d)} - Z^{(d')}||_2^2$**. Therefore, if the loss successfully minimizes this distance such that $Z^{(d)} \approx Z^{(d')}$, it follows necessarily that for any coordinate $k$, the activations are approximately equal: $Z^{(d)}_k \approx Z^{(d')}_k$. Consequently, their expectations (and second moments/energies) must also be equal, satisfying Definition 2.
>
> **Thus, Equation 1 is a geometric proxy that, under the constraint of normalized codes, strictly enforces the Iso-Energy condition at the limit**. We will add a formal derivation in Section 2 to bridge this gap between Definition 2 and Equation 1.
>
> > _"Did the authors experiment with other similar regularizers [...] or other regularizers that can more directly enforce the Iso-Energy property?"_
>
> This is an excellent experimental question. We did indeed explore "direct" implementations of Definition 2.
> Specifically, we experimented with penalties on the explicit difference between activation magnitudes across modalities, using both $L_1$ ($|Z^{(d)} - Z^{(d')}|$) and $L_2$ ($||Z^{(d)} - Z^{(d')}||^2$) norms on the unnormalized activations. **We found these direct energy penalties yielded results indistinguishable from the cosine alignment loss** in terms of dictionary quality across all metrics and interventions. We ultimately selected the cosine formulation (Eq. 1) because it aligns with the native geometry of the SAE decoder and contrastive embedding spaces, which operate on angular similarity. Additionally, since our atoms utilize ReLU-like non-negativity, the "absolute value" variants mentioned in the review are mathematically equivalent to standard inner product maximization in the normalized regime.
>
> > _"The paper introduces the aligned sparse autoencoder without providing sufficient background on the baseline SAE formulation [...]"_
>
> **We agree with this assessment and we apologize for that**. While we initially placed the mathematical formulation of the Matching Pursuit SAE in Appendix A to save space, we recognize that this makes the method section less self-contained.
> **We will move the formal definition of the baseline SAE and its reconstruction/sparsity terms from Appendix A into the main body of Section 2** to ensure the paper is self-contained for readers less familiar with the specific MP-SAE framework.
>
> > _"Some of the mathematical definitions, particularly in Definition 2, are not presented rigorously."_
>
> Good catch, thanks for this correction!
> We have revised Definition 2 to rigorously express the expectation over the domain distribution $X \sim \mathcal{X}^{(d)}$, removing the ambiguous sample-level conditioning.
>
> ---
>
> Thank you again for these constructive criticisms !

---

### Official Review · Reviewer_U7cA · 2025-10-31

**Soundness:** 3
**Presentation:** 4
**Contribution:** 3
**Rating:** 8
**Confidence:** 3

**Summary:**

This paper proposes an Iso-Energy prior for learning aligned sparse concept dictionaries on top of VLM embeddings. By mildly enforcing equal second-moment (“energy”) of a concept across image/text domains, the aligned SAE separates bimodal atoms (semantic carriers) from unimodal atoms (modality-specific bias). This yields two actionable interventions: (i) closing the modality gap by masking unimodal atoms without hurting retrieval, and (ii) performing robust semantic vector arithmetic within the bimodal subspace, reducing OOD drift.

**Strengths:**

1. Clear and effective framing of a testable modeling intuition.
The paper presents a well-motivated and conceptually coherent formulation. It articulates a precise inductive bias: that shared cross-modal concepts should exhibit similar activation statistics across modalities. This idea is not only intuitively appealing but also operationalized in a mathematically minimal way through second-moment constraints. The writing and structural clarity further reinforce this framing, making the contribution accessible and theoretically grounded.

2. Methodologically grounded execution with dual functionality.
The proposed method delivers more than conceptual framing. It constructs a sparse, interpretable bimodal subspace that supports both analysis and intervention. The same subspace allows for attribution-style interpretation as well as semantically coherent editing, demonstrating that the learned structure is not only intelligible but also functionally controllable. This dual capacity is rarely achieved in the interpretability literature and gives the method both analytical and practical value.

**Weaknesses:**

1. Sufficiency versus necessity of the Iso-Energy criterion.
Equal second moments across modalities can indicate shared concepts, but they are not required. Without invariance to modality-specific anisotropy or rescaling, genuinely shared factors may be labeled unimodal. It would be better to add invariance controls such as per-modality whitening or variance normalization, and to compare with covariance-aware baselines such as CCA or CORAL to verify that the findings are not driven by marginal variance.

2. Sensitivity to pairing noise and frequency imbalance.
The alignment term relies on paired image and text data, where long-tail frequencies and noisy matches are common. Energy equality can be confounded by corpus artifacts rather than semantics. It would be better to add two controls: a frequency-matched subsample that balances concept prevalence across modalities, and a shuffled-pairs stress test to quantify robustness to misalignment noise.

**Questions:**

1. To what extent do the conclusions generalize to more complex tasks and architectures, such as VQA on LLaVA-series models?

---

> ### Author Response · Authors · 2025-11-19
> **Response**
>
> **Thank you for your strong endorsement** and for recognizing the “conceptual coherence” and “dual functionality” of our framework. We are particularly grateful for your appreciation of how the method balances theoretical framing with actionable interventions. Below, we address your specific questions.
>
> ---
>
> > _"Sufficiency versus necessity of the Iso-Energy criterion [...] Without invariance to modality-specific anisotropy or rescaling, genuinely shared factors may be labeled unimodal."_
>
> Thank you for this geometric observation. **You are mathematically correct**: strictly speaking, a shared concept could exhibit different variances (anisotropy) across modalities, in which case Iso-Energy might fail to identify it as bimodal.
> However, we argue that in the specific context of contrastive dual-encoders (CLIP, SigLIP), the Iso-Energy assumption is physically motivated by the training objective itself. Specifically we can isolate three factors that motivate our claim
> - **(1) These models are trained to maximize cosine similarity, effectively projecting representations onto the hypersphere**. As noted in Section 2, we process normalized embeddings ($||x||_2 = 1$). **In this regime, magnitude roughly equates to importance**. If a concept is "truly shared" in a contrastive sense, it must contribute to the dot product $\langle x_{img}, x_{txt} \rangle$, implying comparable magnitude in both branches.
> - (2) You suggested whitening or covariance-aware baselines (like CCA). While valid for forcing alignment, our goal was to analyze the geometry as it exists in the pre-trained model. Pre-whitening would artificially collapse the very modality gap we aim to study (Figure 3 1).
> - (3) While perhaps not theoretically necessary (one can imagine shared concepts with different scales), Iso-Energy proves empirically sufficient to recover the alignment backbone. As shown in Figure 2 (Left), the separation between bimodal and unimodal atoms is not a continuum but a distinct bifurcation. That said, we agree this distinction is subtle.
> We will expand Section 2 to explicitly clarify that Iso-Energy is a sufficient inductive bias for practical recovery in contrastive models, rather than a necessary condition for concept existence in general representations.
>
> > _"Sensitivity to pairing noise and frequency imbalance [...] Energy equality can be confounded by corpus artifacts."_
>
> This is a valid concern given the noisy nature of web-scale datasets like LAION. We address this through two mechanisms: first, **Iso-Energy not as a hard constraint, but as a soft penalty** ($\beta \cdot \mathcal{L}_{align}$, Eq. 1 3). This allows the model to tolerate instance-level noise (e.g., bad pairings) provided the concept is shared on average across the corpus. Then **our dictionary is learned over 1 million embeddings. While individual pairs may be noisy or mismatched, the "bimodal backbone" emerges from the aggregate statistics of the dataset**. The consistency of our results across 6 different models (Table 1 4) suggests the discovered structure is robust to specific corpus artifacts.
> However, we agree that frequency imbalance (long-tail concepts) poses a challenge. A concept appearing frequently in text but rarely in images might be penalized. We will add a discussion in the Limitations section regarding frequency imbalance and suggest frequency-matched controls as a standard for future rigorous validation.
>
> > _"To what extent do the conclusions generalize to more complex tasks and architectures, such as VQA on LLaVA-series models?"_
>
> **This is an excellent question**. Honestly, the extension to VQA models (like LLaVA) is non-trivial.Our current formalism relies on the symmetry of the dual-encoder embedding space. In contrast, LLaVA-style architectures project visual tokens into a frozen LLM space. In that setting, "alignment" is not geometric congruence (as in CLIP) but rather functional translation as the visual tokens must mimic the statistics of text tokens to drive generation. We hypothesize that a variant of Iso-Energy could still apply to the projector outputs, but the definition of "energy" might need to account for the causal attention mechanism of the LLM. We view this as a really exciting direction for future work and will expand the discussion (Section 5) to explicitly contrast the dual-encoder geometry with the projection-based geometry of VQA models.
>
> ---
>
> Thank you again for your constructive feedback. We believe incorporating these clarifications will significantly strengthen the manuscript.

---

### Meta-Review · Area_Chair_QffE · 2026-01-07

**Summary:**

The rebuttal provides substantial clarification regarding the mathematical link between the proposed assumption and the alignment loss, and it significantly strengthens the related-work discussion—particularly in response to Reviewer XysU’s concerns. Overall, the authors do a good job clarifying and differentiating their contributions from prior work. In addition, Reviewer BMn2 raised questions about the practical scope and applicability, which the authors also addressed. For these reasons, I recommend acceptance.

**Reviewer Concerns:**

The authors did a solid job addressing concerns about the theory-to-implementation link (Iso-Energy vs. Eq. (1)), clarifying key methodological details, and situating their contributions relative to prior work. A remaining outstanding concern is robustness to pairing noise and frequency imbalance: while the rebuttal acknowledges these issues, it does not fully address performance and reliability in long-tailed settings.

**Reviewer Scores:**

I think Reviewer U7cA may keep the scores and Reviewer Yfxj may raise the score to 6, the reviewer BMn2 may keeps it and not sure about the reviewer XysU, I acknowledge the authors clarification on the difference discussion to the previous work.

---

### Decision · Program_Chairs · 2026-01-26

Accept (Poster)